# ESTIMATING CONDITIONAL MUTUAL INFORMATION FOR DYNAMIC FEATURE SELECTION

**Soham Gadgil**[*], **Ian Covert**[*], **Su-In Lee**
Paul G. Allen School of Computer Science & Engineering, University of Washington

## ABSTRACT

Dynamic feature selection, where we sequentially query features to make accurate predictions with a minimal budget, is a promising paradigm to reduce feature acquisition costs and provide transparency into a model's predictions. The problem is challenging, however, as it requires both predicting with arbitrary feature sets and learning a policy to identify valuable selections. Here, we take an information-theoretic perspective and prioritize features based on their mutual information with the response variable. The main challenge is implementing this policy, and we design a new approach that estimates the mutual information in a discriminative rather than generative fashion. Building on our approach, we then introduce several further improvements: allowing variable feature budgets across samples, enabling non-uniform feature costs, incorporating prior information, and exploring modern architectures to handle partial inputs. Our experiments show that our method provides consistent gains over recent methods across a variety of datasets.

## 1 INTRODUCTION

Many machine learning applications rely on high-dimensional datasets with significant data acquisition costs. For example, medical diagnosis can depend on a range of demographic features, lab tests and physical examinations, and each piece of information takes time and money to collect (Kachuee et al., 2018; Erion et al., 2022; He et al., 2022). To improve interpretability and reduce data acquisition costs, a natural approach is to adaptively query features given the current information, so that each prediction relies on only a small number of features. This approach is referred to as *dynamic feature selection* (DFS),[1] and it is a promising paradigm considered by several works in recent years (Kachuee et al., 2018; Janisch et al., 2019; Chattopadhyay et al., 2022; 2023; Covert et al., 2023).

Among the existing methods that address this problem, two main approaches have emerged. One idea is to formulate DFS as a Markov decision process (MDP) and use reinforcement learning (RL) (Dulac-Arnold et al., 2011; Mnih et al., 2014; Kachuee et al., 2018; Janisch et al., 2019). This approach theoretically has the capacity to discover the optimal policy, but it faces training difficulties that are common in RL (Henderson et al., 2018). Alternatively, another line of work focuses on greedy approaches, where features are selected based on their conditional mutual information (CMI) with the response variable (Chen et al., 2015; Ma et al., 2019). While less flexible than RL, the greedy approach is near-optimal under certain assumptions about the data distribution (Chen et al., 2015) and represents a simpler learning problem; as a result, it has been found to perform better than RL in several recent works (Erion et al., 2022; Chattopadhyay et al., 2023; Covert et al., 2023).

Nevertheless, the greedy approach is non-trivial to implement, because calculating the CMI requires detailed knowledge of the data distribution. Many recent works have explored approximating the CMI using generative models (Ma et al., 2019; Rangrej and Clark, 2021; Chattopadhyay et al., 2022; He et al., 2022), but these are difficult to train and lead to a slow CMI estimation process. Instead, two recent works introduced a simpler approach, which is to directly estimate the feature index with maximum CMI (Chattopadhyay et al., 2023; Covert et al., 2023). These methods rely on simpler learning objectives, are faster at inference time, and often provide better predictive accuracy. They can

---

[*]Equal contribution. Correspondence to <sgadgil@cs.washington.edu>.
[1]Prior works have also referred to the problem as as *sequential information maximization* (Chen et al., 2015), *active variable selection* (Ma et al., 2019) and *information pursuit* (Chattopadhyay et al., 2023).

be thought of *discriminative* alternatives to earlier generative methods, because they directly output the required value rather than modeling the data generation process (Ng and Jordan, 2001). Their main downside is that they bypass estimating the CMI, which can be useful for multiple purposes, such as determining when to stop selecting new features.

Here, our goal is to advance the greedy DFS approach by combining the best aspects of current methods: building on recent work (Chattopadhyay et al., 2023; Covert et al., 2023), we aim to *estimate the CMI itself* in a discriminative fashion. We aim to do so without requiring additional labels, making strong assumptions about the data distribution, or fitting generative models. We accomplish this by designing a suitable learning objective, which we prove recovers the CMI if our model is trained to optimality (Section 4).

Based on our new learning approach, we then explore a range of capabilities enabled by accurately estimating the CMI: these include allowing variable per-sample feature budgets, accounting for non-uniform feature costs, and leveraging modern architectures to train models with partial input information. In real-world settings, acquisition costs are not identical for all features, and we can use multiple low-cost features in lieu of one high-cost feature to get the same diagnostic performance. Although these capabilities are supported by certain RL methods (Kachuee et al., 2018; Janisch et al., 2019), they are not supported by prior discriminative approaches that do not estimate the CMI.

We find that our proposal offers a promising alternative to available methods: it enables the capabilities of generative methods while retaining the simplicity of discriminative methods, and it shows improved performance in our experiments. The contributions of this work are the following:

1. We develop a learning approach to estimate the CMI in a discriminative fashion. Our method involves training a network to score candidate features based on their predictive utility, and we prove that training with our objective recovers the exact CMI at optimality.

2. We generalize our approach to incorporate prior information beyond the main features. Here, we again prove that our procedure recovers a modified version of the CMI at optimality.

3. Taking inspiration from adaptive submodular optimization, we show how to adapt our CMI-based approach to scenarios with non-uniform feature costs.

4. We analyze the role of variable feature budgets and how they enable an improved cost-accuracy tradeoff. We show that a single instantiation of our method can be evaluated with multiple stopping criteria, and that a policy with a flexible per-prediction budget performs best.

5. We investigate the role of modern architectures in improving performance in DFS. In particular, we find that for image data, our method benefits from using ViTs rather than standard CNNs.

Our experiments demonstrate the effectiveness of our approach across a range of applications, including several tabular and image datasets. We compare our approach to many recent methods, and we find that our approach provides consistent gains across all the datasets we tested.

## 2 PROBLEM FORMULATION

**Notation.** Let $\mathbf{x}$ denote a vector of input features and $\mathbf{y}$ a response variable for a supervised learning task. The input consists of $d$ separate features $\mathbf{x} = (\mathbf{x}_1, \ldots, \mathbf{x}_d)$, and we use $S \subseteq [d] \equiv \{1, \ldots, d\}$ to denote a subset of indices and $\mathbf{x}_S = \{\mathbf{x}_i : i \in S\}$ a subset of features. Bold symbols $\mathbf{x}, \mathbf{y}$ represent random variables, the symbols $x, y$ are possible values, and $p(\mathbf{x}, \mathbf{y})$ denotes the data distribution.

Our goal is to select features given the currently available information, and do so on a per-instance basis to rapidly arrive at accurate predictions. In doing so, we require a predictor $f(\mathbf{x}_S)$ that makes predictions given any set of available features; for example, if $\mathbf{y}$ is discrete then our predictions lie in the simplex, or $f(\mathbf{x}_S) \in \Delta^{K-1}$ for $K$ classes. We also require a selection policy $\pi(\mathbf{x}_S) \in [d]$, which takes a set of features as its input and outputs the next feature index to observe. We next discuss how to design these models, focusing on an approach motivated by information theory.

**Dynamic feature selection.** The goal of DFS is to select features separately for each prediction, and achieve both low acquisition cost and high predictive accuracy. Previous work has explored several approaches to design a selection policy, including training the policy with RL (Kachuee et al., 2018; Janisch et al., 2019), imitation learning (He et al., 2012; 2016a), and following a greedy

policy based on CMI (Ma et al., 2019; Covert et al., 2023). We focus here on the latter approach, where the idealized policy selects the feature with maximum CMI at each step, or where we define $\pi^*(x_S) = \arg\max_i I(\mathbf{y}; \mathbf{x}_i \mid x_S)$. Intuitively, the CMI represents how much observing $\mathbf{x}_i$ improves our ability to predict $\mathbf{y}$ when a subset of the features $x_S$ is already observed, and it is defined as the following KL divergence (Cover and Thomas, 2012):

$$I(\mathbf{y}; \mathbf{x}_i \mid x_S) = D_{\mathrm{KL}}\left(p(\mathbf{x}_i, \mathbf{y} \mid x_S) \,\|\, p(\mathbf{x}_i \mid x_S)p(\mathbf{y} \mid x_S)\right). \tag{1}$$

The idealized selection policy is accompanied by an idealized predictor, which is the Bayes classifier $f^*(x_S) = p(\mathbf{y} \mid x_S)$ for classification problems, and under certain assumptions it is known that this provides performance within a multiplicative factor of the optimal policy (Chen et al., 2015). However, the CMI policy is difficult to implement because it requires oracle access to the data distribution: computing eq. (1) requires both the distributions $p(\mathbf{y} \mid x_S)$ and $p(\mathbf{x}_i \mid x_S)$ for all $(S, i)$, which presents a challenging modeling problem. For example, some works have approximated $I(\mathbf{y}; \mathbf{x}_i \mid x_S)$ using generative models (Ma et al., 2019; Rangrej and Clark, 2021; He et al., 2022), while others have directly modeled $\pi^*(x_S)$ (Chattopadhyay et al., 2023; Covert et al., 2023).

When we follow the greedy CMI policy, another question that arises is how many features to select for each prediction. Previous work has focused mainly on the fixed-budget setting, where we stop when $|S| = k$ for a specified budget $k < d$ (Ma et al., 2019; Rangrej and Clark, 2021; Chattopadhyay et al., 2023; Covert et al., 2023). We instead consider variable budgets in this work (Kachuee et al., 2018; Janisch et al., 2019), where the goal is to achieve high accuracy given a low *average feature cost*. Unlike many works, we also consider distinct, or non-uniform costs for each feature. As we discuss in Section 4, these goals are made easier by estimating the CMI in a discriminative fashion.

## 3 RELATED WORK

One of the earliest works on DFS is Geman and Jedynak (1996), who used the CMI as a selection criterion but made simplifying assumptions about the data distribution. Chen et al. (2015) analyzed the greedy CMI approach theoretically and proved conditions where it achieves near-optimal performance. More recent works have focused on practical implementations: among them, several use generative models to approximate the CMI (Ma et al., 2019; Rangrej and Clark, 2021; Chattopadhyay et al., 2022; He et al., 2022), and two works proposed predicting the best feature index in a discriminative fashion (Chattopadhyay et al., 2023; Covert et al., 2023). Our work develops a similar discriminative approach, but to estimate the CMI itself rather than the argmax among candidate features.

Apart from these CMI-based methods, other works have addressed DFS as an RL problem (Dulac-Arnold et al., 2011; Shim et al., 2018; Kachuee et al., 2018; Janisch et al., 2019; 2020; Li and Oliva, 2021). For example, Janisch et al. (2019) formulate DFS as a MDP where the reward is the 0-1 loss minus the feature cost, while considering both variable budgets and non-uniform feature costs. RL theoretically has the capacity to discover better policies than a greedy approach, but it has not been found to perform well in practice, seemingly due to training difficulties that are common in RL (Erion et al., 2022; Chattopadhyay et al., 2023; Covert et al., 2023). Beyond these approaches, other works have instead explored imitation learning by mimicking an oracle policy (He et al., 2012; 2016a), minimizing expected misclassification costs at each step (Liyanage et al., 2021b), and making selections based on a Bayesian network fit to the joint data distribution (Liyanage et al., 2021a).

Static feature selection has been an important subject in statistics and machine learning for decades; see (Guyon and Elisseeff, 2003; Li et al., 2017; Cai et al., 2018) for reviews. CMI is also the basis of some static methods, and its estimation has been studied by many works (Fleuret, 2004; Peng et al., 2005; Shishkin et al., 2016). Greedy methods perform well under certain assumptions about the data distribution and are popular for simple models (Das and Kempe, 2011; Elenberg et al., 2018), but feature selection with nonlinear models is more challenging. For neural networks, methods now exist that leverage either group sparse penalties (Feng and Simon, 2017; Tank et al., 2021; Lemhadri et al., 2021) or differentiable gating mechanisms (Chang et al., 2017; Balın et al., 2019; Lindenbaum et al., 2021). Unlike recent DFS methods (Chattopadhyay et al., 2023; Covert et al., 2023), our work bypasses these techniques by using a simpler regression objective.

Finally, mutual information estimation with deep learning has been an active topic in recent years (Oord et al., 2018; Belghazi et al., 2018; Poole et al., 2019; Song and Ermon, 2019; Shalev et al., 2022). Unlike prior works, ours focuses on the CMI between features and a response variable,

our method can condition on arbitrary feature sets, and we estimate many CMI terms via a single predictive model rather than training separate networks to estimate each mutual information term.

# 4 PROPOSED METHOD

In this section, we introduce our method to dynamically select features by estimating the CMI in a discriminative fashion. We then discuss how to incorporate prior information into the selection process, handle non-uniform feature costs, and enable variable budgets across predictions.

## 4.1 ESTIMATING THE CONDITIONAL MUTUAL INFORMATION

We parameterize two networks to implement our selection policy. First, we have a predictor network $f(\mathbf{x}_S; \theta)$, e.g., a classifier with predictions in $\Delta^{K-1}$. Next, we have a value network $v(\mathbf{x}_S; \phi) \in \mathbb{R}^d$ designed to estimate the CMI for each feature, or $v_i(x_S; \phi) \approx I(\mathbf{y}; \mathbf{x}_i \mid x_S)$. These are implemented with zero-masking to represent missing features, and we can also pass the mask as a binary indicator vector. Once they are trained, we make selections according to $\arg\max_i v_i(\mathbf{x}_S; \phi)$, and we can make predictions at any time using $f(\mathbf{x}_S; \theta)$. The question is how to train the networks, given that prior works required generative models to estimate the CMI (Ma et al., 2019; Chattopadhyay et al., 2022).

Our main insight is to train the models jointly but with their own objectives, and to design an objective for the value network that recovers the CMI at optimality. Specifically, we formulate a regression problem whose target is the incremental improvement in the loss when incorporating a single new feature. We train the predictor to make accurate predictions given any feature set, or

$$\min_\theta \ \mathbb{E}_{\mathbf{xy}} \mathbb{E}_{\mathbf{s}} \left[ \ell \left( f(\mathbf{x}_\mathbf{s}; \theta), \mathbf{y} \right) \right], \tag{2}$$

and we simultaneously train the value network with the following regression objective,

$$\min_\phi \ \mathbb{E}_{\mathbf{xy}} \mathbb{E}_{\mathbf{s}} \mathbb{E}_i \left[ \left( v_i(\mathbf{x}_\mathbf{s}; \phi) - \Delta(\mathbf{x}_\mathbf{s}, \mathbf{x}_i, \mathbf{y}) \right)^2 \right], \tag{3}$$

where we define the loss improvement as $\Delta(x_S, x_i, y) = \ell(f(x_S; \theta), y) - \ell(f(x_{S \cup i}; \theta), y)$. The regression objective in eq. (3) is motivated by the following property, which shows that if we assume an accurate predictor $f(\mathbf{x}_S; \theta)$ (i.e., the Bayes classifier), the value network's labels are unbiased estimates of the CMI (proofs are in Appendix A).

**Lemma 1.** *When we use the Bayes classifier $p(\mathbf{y} \mid \mathbf{x}_S)$ as a predictor and $\ell$ is cross entropy loss, the incremental loss improvement is an unbiased estimator of the CMI for each $(x_S, \mathbf{x}_i)$ pair:*

$$\mathbb{E}_{\mathbf{y}, \mathbf{x}_i \mid x_S} \left[ \Delta(x_S, \mathbf{x}_i, \mathbf{y}) \right] = I(\mathbf{y}; \mathbf{x}_i \mid x_S). \tag{4}$$

Based on this result and the fact that the optimal predictor does not depend on the selection policy (Covert et al., 2023), we can make the following claim about jointly training the two models. We assume that both models are infinitely expressive (e.g., very wide networks) so that they can achieve their respective optimizers.

**Theorem 1.** *When $\ell$ is cross entropy loss, the objectives eq. (2) and eq. (3) are jointly optimized by a predictor $f(x_S; \theta^*) = p(\mathbf{y} \mid x_S)$ and value network where $v_i(x_S; \phi^*) = I(\mathbf{y}; \mathbf{x}_i \mid x_S)$ for $i \in [d]$.*

This allows us to train the models in an end-to-end fashion using stochastic gradient descent, as depicted in Figure 1. In Appendix A we prove a similar result for regression problems: that the policy estimates the reduction in conditional variance associated with each candidate feature. Additional analysis in Appendix B shows how suboptimality in the classifier can affect the learned CMI estimates; however, even if the learned estimates $v_i(\mathbf{x}_S; \phi)$ are imperfect in practice, we expect good performance because the policy replicates selections that improve the loss during training.

Several other steps are important during training, and these are detailed in Appendix C along with pseudo-code for the training algorithm and the inference procedure. First, like several prior methods, we pre-train the predictor with random feature sets before beginning joint training (Rangrej and Clark, 2021; Chattopadhyay et al., 2023; Covert et al., 2023). Next, we generate training samples $(x_S, x_i, y)$ by executing the current policy with a random exploration probability $\epsilon \in [0, 1]$, which can be decayed throughout training. Finally, we sometimes share parameters between the models, particularly when they are large networks; this helps in our experiments with image data, which use either CNNs or ViTs (Dosovitskiy et al., 2020).

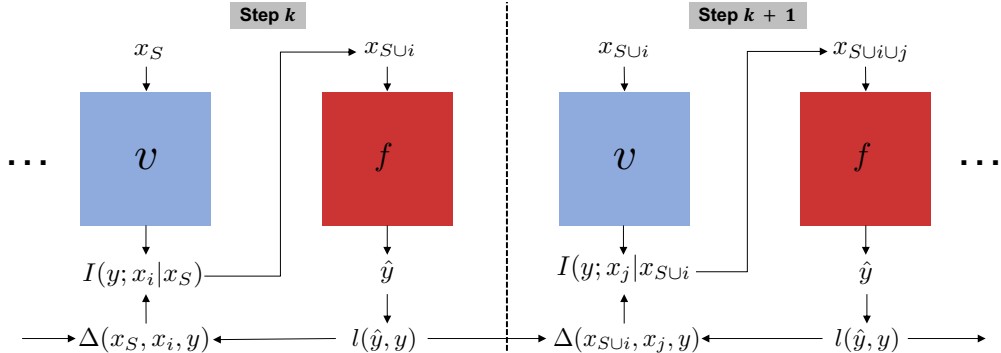

Figure 1: Diagram of our training approach. At each selection step $n$, the value network $v(x_S; \phi)$ predicts the CMI for all features, and a single feature $x_i$ is chosen for the next prediction $f(x_{S\cup i}; \theta)$. The prediction loss is used to update the predictor (see eq. (2)), and the loss improvement is used to update the value network (see eq. (3)). The networks are trained jointly with SGD.

## 4.2   INCORPORATING PRIOR INFORMATION

A further direction is utilizing *prior information* obtained before beginning the selection process. We view such prior information as a separate random variable $\mathbf{z}$, and it can be either an exogenous input, a subset of features that are available with no associated cost, or even a noisy or low-resolution version of the main input $\mathbf{x}$ (Ranzato, 2014; Ba et al., 2014; 2015). Situations of this form arise in multiple applications, e.g., a patient's demographic features in a medical diagnosis setting.

Given such prior information, our idealized DFS policy must be modified as follows. First, the selections should be based on $I(\mathbf{y}; \mathbf{x}_i \mid x_S, z)$, which captures how informative $\mathbf{x}_i$ is given knowledge of both $\mathbf{x}_S$ and $\mathbf{z}$. Next, the predictions made at any time are given by $p(\mathbf{y} \mid x_S, z)$, because $\mathbf{z}$ provides information that can improve our predictions. As for our proposed CMI estimation procedure from Section 4.1, it is straightforward to modify. The two models must take the prior information $\mathbf{z}$ as an additional input, and we can train them with modified versions of eqs. (2) and (3),

$$\min_{\theta} \ \mathbb{E}_{\mathbf{xyz}}\mathbb{E}_{\mathbf{s}}\left[\ell\left(f(\mathbf{x}_{\mathbf{s}}, \mathbf{z}; \theta), y\right)\right], \qquad \min_{\phi} \ \mathbb{E}_{\mathbf{xyz}}\mathbb{E}_{\mathbf{s}}\mathbb{E}_i\left[\left(v_i(\mathbf{x}_{\mathbf{s}}, \mathbf{z}; \phi) - \Delta(\mathbf{x}_{\mathbf{s}}, \mathbf{x}_i, \mathbf{z}, \mathbf{y})\right)^2\right], \quad (5)$$

where $\Delta(x_S, x_i, z, y) = \ell(f(x_S, z, ; \theta), y) - \ell(f(x_{S\cup i}, z; \theta), y)$ is the incremental loss improvement. We then have the following result for jointly training the two models.

**Theorem 2.** *When $\ell$ is cross entropy loss, the objectives in eq.* (5) *are jointly optimized by a predictor* $f(x_S, z; \theta^*) = p(\mathbf{y} \mid x_S, z)$ *and value network where* $v_i(x_S, z; \phi^*) = I(\mathbf{y}; \mathbf{x}_i \mid x_S, z)$ *for all $i \in [d]$.*

The same implementation details discussed in Section 4.1 apply here, and Section 5.2 discusses how the prior information $\mathbf{z}$ is incorporated into the value and predictor networks in practice.

## 4.3   ALLOWING A VARIABLE FEATURE BUDGET

Given our approach for estimating each feature's CMI, a natural question is how to trade off information with feature acquisition costs. We now consider two challenges related to features costs: (1) how to handle non-uniform costs between features, and (2) when to stop collecting new features.

**Non-uniform costs.** For the first challenge, consider medical diagnosis as a motivating example. Diagnoses can be informed by heterogeneous data, including demographic variables, questionnaires, physical examinations, and lab tests; each measurement requires a different amount of time or money (Kachuee et al., 2018; Erion et al., 2022), and feature costs must be balanced with the information they provide. We consider here that each feature has a cost $c_i > 0$ and that costs are additive.

There are multiple ways to trade off cost with information, but we take inspiration from adaptive submodular optimization, where costs are accounted for via the ratio between the expected improvement and cost. Here, this suggests that our selections should be $\arg\max_i I(\mathbf{y}; \mathbf{x}_i \mid x_S)/c_i$. For adaptive submodular objectives, this criterion guarantees near-optimal performance (Golovin and Krause, 2011); the DFS problem is known to *not* be adaptive submodular (Chen et al., 2015), which means that we cannot offer performance guarantees, but we find that this approach works well in practice.

**Variable budgets.** Next, we consider when to stop acquiring new features. Many previous works focused on the budget-constrained setting, where we adopt a budget $k$ for all predictions (Chen et al., 2015; Ma et al., 2019; Rangrej and Clark, 2021; Covert et al., 2023). This can be viewed as a stopping criterion, and it generalizes to non-uniform costs: we can keep collecting new information as long as $\sum_{i \in S} c_i \leq k$. Alternatively, we can adopt a confidence-constrained setup (Chattopadhyay et al., 2023), where selection terminates once the predictions have low uncertainty. For classification problems, a natural approach is to stop collecting features when $H(\mathbf{y} \mid x_S) \leq m$.

In general, it is unclear whether we should follow a budget- or confidence-constrained approach, or whether another option offers a better cost-accuracy tradeoff. We resolve this by considering the optimal performance achievable by *non-greedy policies*, and we present the following insight: that policies with per-prediction constraints are Pareto-dominated by those that satisfy their constraints *on average*. We state this claim here in a simplified form, and we defer the formal version to Appendix A.

**Proposition 1.** *(Informal) For any feature budget $k$, the best policy to achieve this budget on average achieves lower loss than the best policy with a per-prediction budget constraint. Similarly, for any confidence level $m$, the best policy to achieve this confidence on average achieves lower cost than the best policy with a per-prediction confidence constraint.*

Intuitively, when designing a policy to achieve a given average cost or confidence level, it should help to let the policy violate that level for certain predictions. For example, for a patient whose medical condition is inherently uncertain and will not be resolved by any number of tests, it is preferable from a cost-accuracy perspective to stop early rather than run many expensive tests. Proposition 1 suggests that we should avoid adopting budget or confidence constraints and instead seek the optimal unconstrained policy, but because we assume that we only have CMI estimates (Section 4.1), we opt for a simple alternative: we adopt a penalty parameter $\lambda > 0$, we make selections at each step according to $I(\mathbf{y}; \mathbf{x}_i \mid x_S)/c_i$, and we terminate the algorithm when $\max_i I(\mathbf{y}; \mathbf{x}_i \mid x_S)/c_i < \lambda$.

Following this approach, we see that a single instantiation of our model can be run with three different stopping criteria: we can use the budget $k$, the confidence $m$, or the penalty $\lambda$. In contrast, prior methods that penalize feature costs required training separately for each penalty strength (Janisch et al., 2019). Although our penalized criterion is not the optimal one alluded to in Proposition 1, we find empirically that it consistently improves performance relative to a fixed budget.

## 5 EXPERIMENTS

We refer to our approach as *DIME*[2] (**di**scriminative **m**utual information **e**stimation) and explore two data modalities to evaluate its performance. Our tabular datasets include two medical diagnosis tasks, which represent natural and valuable use cases for DFS; we also use MNIST, which was considered in prior works (Ma et al., 2019; Chattopadhyay et al., 2022; 2023). As for our image datasets, we include these because they are studied in several earlier works and represent challenging DFS problems (Karayev et al., 2012; Mnih and Gregor, 2014; Janisch et al., 2019; Rangrej and Clark, 2021). We also design an experiment to test the CMI estimation accuracy in Appendix G.

In terms of baselines, we compare DIME to both static and dynamic feature selection methods. As a strong static baseline, we compare to a supervised Concrete Autoencoder (*CAE*) (Balın et al., 2019), which outperformed several dynamic methods in recent work (Covert et al., 2023). We also compare to two more static baselines that statistically estimate the CMI for feature selection, *mRMR* (Peng et al., 2005) and *CMICOT* (Shishkin et al., 2016). For dynamic baselines, we consider multiple approaches: first, we compare to the recent discriminative methods that directly predict the CMI's argmax (Chattopadhyay et al., 2023; Covert et al., 2023), which we refer to as *Argmax Direct*. Next, we consider *EDDI* (Ma et al., 2019), a generative approach that uses a partial variational autoencoder (PVAE) to sample unknown features. Because EDDI does not scale as well beyond tabular datasets, we compare to probabilistic hard attention (*Hard Attention*) (Rangrej and Clark, 2021), a method that adapts EDDI to work with image data. Finally, we also compare the tabular datasets to two RL approaches: classification with costly features (*CwCF*) (Janisch et al., 2020) and opportunistic learning (*OL*) (Kachuee et al., 2018), which both design rewards based on features' predictive utility. Appendix F provides more information about the baseline methods.

---

[2]Code is available at `https://github.com/suinleelab/DIME`

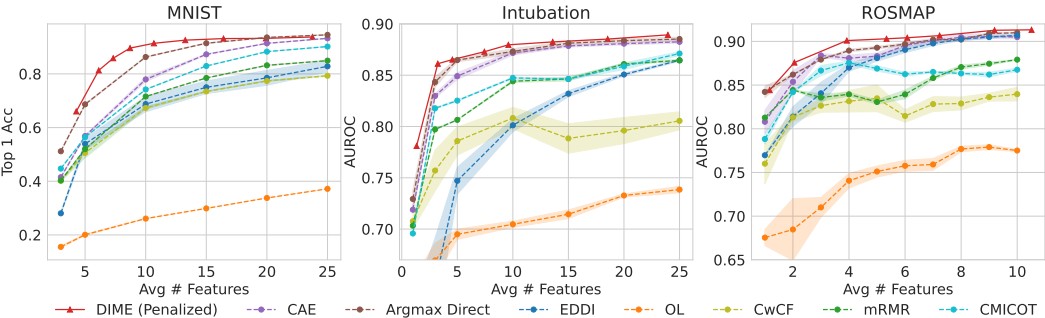

Figure 2: Evaluation with tabular datasets for varying feature acquisition budgets. Results are averaged across 5 trials, and shaded regions indicate the standard error for each method.

## 5.1 TABULAR DATASETS

Our first medical diagnosis task involves predicting whether a patient requires endotracheal intubation for respiratory support in an emergency medicine setting ($d = 112$) (Covert et al., 2023). The second uses cognitive, demographic, and medical history data from two longitudinal aging cohort studies (ROSMAP, A Bennett et al. 2012a;b) to predict imminent dementia onset ($d = 46$). In both scenarios, it is difficult to acquire all features due to time constraints, making DFS a promising paradigm. The third is the standard MNIST dataset (LeCun et al., 1998), which we formulate as a tabular problem by treating each pixel as a feature ($d = 784$). Across all methods, we use fully connected networks with dropout to reduce overfitting (Srivastava et al., 2014), and the classification performance is measured using AUROC for the medical tasks and top-1 accuracy for MNIST. Appendix D provides more details about the datasets, and Appendix E provides more information about the models.

**Uniform feature costs.** We first consider the scenario with equal costs for all features, and Figure 2 shows results with a range of feature budgets. DIME with the penalized stopping criterion achieves the best results among all methods for both medical diagnosis tasks. It also performs the best on MNIST, where we achieve above $90\%$ accuracy with an average of $\sim 10/784$ features ($1.27\%$). Among the baselines, Argmax Direct is the strongest dynamic method, CAE is a competitive static method that outperforms both CMICOT and mRMR, and EDDI usually does not perform well. The RL approaches (CwCF, OL) are not as competitive, as expected from results in prior work (Rangrej and Clark, 2021; Chattopadhyay et al., 2023; Covert et al., 2023). DIME generally shows the greatest advantage for moderate numbers of features, and the gap reduces as the performance saturates. The variability between trials for the baselines is typically low, as shown by our uncertainty estimates; for DIME, we show results from five independent trials in Appendix G because it is difficult to ensure identical budgets with separate training runs.

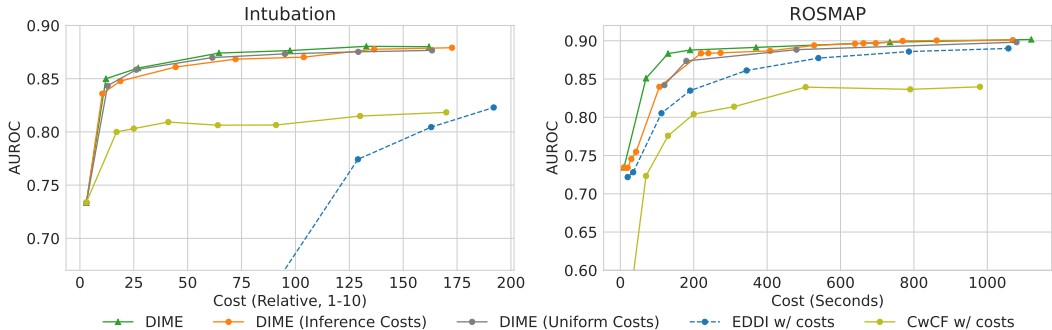

Figure 3: Evaluation with non-uniform feature costs for medical diagnosis tasks. Costs are relative for Intubation and expressed in seconds for ROSMAP. The results show the classification performance for varying levels of average feature acquisition cost.

**Non-uniform feature costs.** Our CMI estimation approach lets us incorporate non-uniform feature costs into DIME, and we demonstrate this with the Intubation and ROSMAP datasets. For ROSMAP, we use costs expressed as the time required to acquire each feature, and for Intubation we use relative costs estimated by a board-certified physician (Appendix D). For comparisons, we use EDDI and

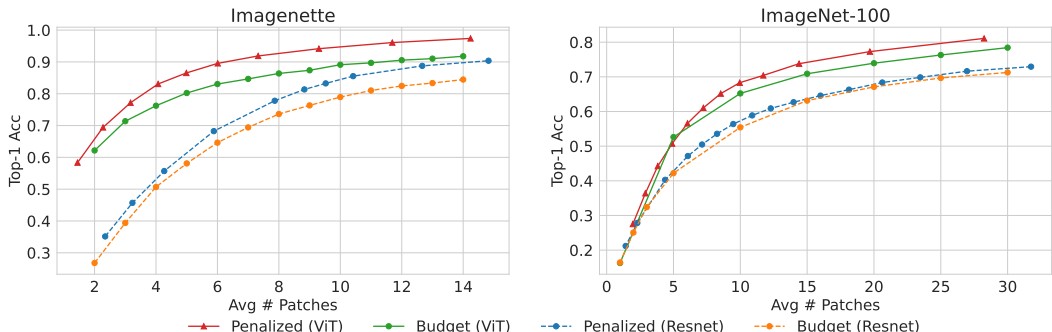

Figure 4: Evaluation of DIME on image datasets with different vision architectures.

CwCF to accommodate non-uniform feature costs. We also compare to two ablations of our approach: (1) using uniform costs during training but the true costs during inference (Inference Costs), which tests DIME's robustness to changing feature costs after training; and (2) using uniform costs during both training and inference (Uniform Costs), which demonstrates the importance of using correct costs. All methods are compared here with the budget-constrained stopping criterion.

Figure 3 shows the results with non-uniform feature costs. DIME outperforms EDDI and CwCF by a substantial margin, echoing the earlier results, and reflecting the improved CMI estimation with our discriminative approach. Comparing to the variations of DIME, using the true non-uniform costs during both training and inference outperforms both variations, showing that considering costs when making selections is important. The version that uses costs only during inference slightly outperforms ignoring costs on ROSMAP, indicating a degree of robustness to changing feature costs between training and inference, but both versions give similar results on Intubation.

**Additional results.** Due to space constraints, we defer several additional results to Appendix G.

## 5.2 IMAGE DATASETS

Next, we applied our method to three image classification datasets. The first two are subsets of ImageNet (Deng et al., 2009), one with 10 classes (Imagenette, Howard) and the other with 100 classes (ImageNet-100, Ambityga). The third is a histopathology dataset (MHIST, Wei et al. 2021), comprising hematoxylin and eosin (H&E)-stained images obtained by extracting diagnostically-relevant tiles from whole-slide images (WSIs). The task is to predict the histological pattern as either a benign or precancerous lesion. WSIs have extremely high resolution and are infeasible for direct use in any classification task, making them a potential use case for DIME to identify important patches. The images in all three datasets are $224 \times 224$, and we view them as $d = 196$ patches of size $16 \times 16$. We explore different architectures for the value and predictor networks, namely ResNets (He et al., 2016b) and Vision Transformers (ViTs) (Dosovitskiy et al., 2020). We use a shared backbone in both cases, with each component having its own output head. Classification performance is measured using top-1 accuracy for both ImageNet subsets, and with AUROC for the histopathology task.

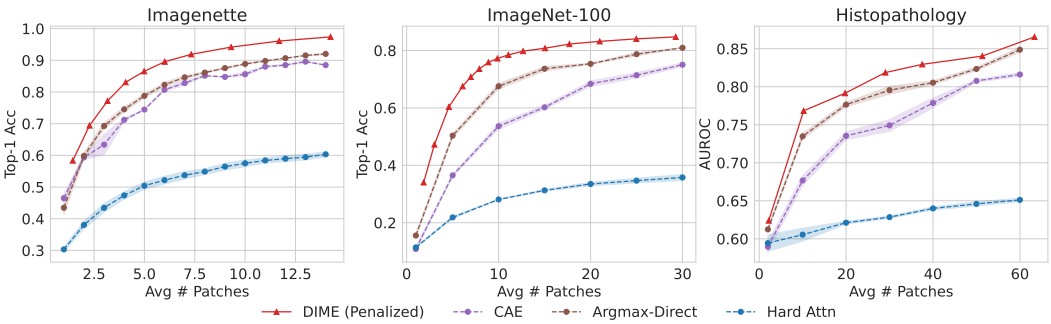

Figure 5: Evaluation with image datasets for varying numbers of average patches selected. Results are averaged across 5 trials, and shaded regions indicate the standard error for each method.

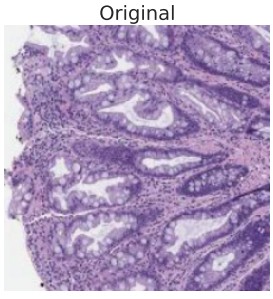 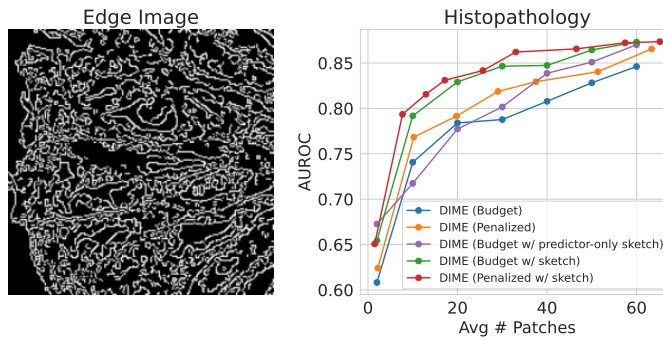

Figure 6: Evaluation of DIME with prior information for histopathology classification. Left: Original MHIST image. Center: Canny edge image. Right: Results for varying number of average patches.

**Exploring modern architectures.** As an initial experiment, Figure 4 compares the ResNet and ViT architectures for the Imagenette and ImageNet-100 datasets. We use DIME to conduct this analysis because its discriminative approach lets us seamlessly plug in any network architecture. Across both datasets, DIME's penalized version outperforms the budget-constrained version, and we find that ViTs significantly outperform ResNets. This can be attributed to the ViT's self-attention mechanism: this architecture is better suited to handle information spread across an image, a property that has made ViTs useful in other applications with partial inputs (Naseer et al., 2021; Jain et al., 2021; Salman et al., 2022). Given their superior performance, we use ViTs as backbones in subsequent experiments where possible. This includes DIME, CAE and Argmax Direct, but Hard Attention lacks this flexibility because it uses a recurrent module to process subsets of image regions.

Next, Figure 5 compares DIME to the baselines across multiple feature budgets. DIME with the penalized stopping criteria outperforms the baselines for all feature budgets, with the largest gains observed for Imagenette. Notably, we achieve nearly 97% accuracy on Imagenette with only ~15/196 patches (7.7%). The Argmax Direct baseline is competitive with DIME, but the Hard Attention baseline shows a larger drop in performance.

**Incorporating prior information.** Next, we explored the possibility of incorporating prior information into DIME's selection process for the histopathology dataset. To simulate informing our selections with a less exact but easily acquirable version of the tissue, we use the Canny edge image as a sketch (Canny, 1986), which can help generate more valuable selections than a blank image. We use separate ViT backbones for the original and edge images, and we concatenate the resulting embeddings before estimating the CMI or making class predictions (see Appendix E for details). Figure 6 shows example images, along with the results obtained with DIME for various feature budgets. The results show that the prior information is incorporated successfully, leading to improved performance for both the penalized and budget-constrained versions. To verify that the improvement is not solely due to the predictive signal provided by the edge image, we conduct an ablation where the sketch is integrated into the predictor only for a pre-trained and otherwise frozen version of DIME. This middle ground improves upon no prior information, but it generally performs well below the version that uses prior information both when making selections and predictions.

## 6 CONCLUSION

This work presents DIME, a new DFS approach enabled by estimating the CMI in a discriminative fashion. Our approach involves learning value and predictor networks, trained in an end-to-end fashion with a straightforward regression objective. From a theoretical perspective, we prove that our training approach recovers the exact CMI at optimality. Empirically, DIME accurately estimates the CMI and enables an improved cost-accuracy tradeoff, exceeding both the generative and discriminative methods it builds upon (Chattopadhyay et al., 2023; Covert et al., 2023). Our evaluation considers a range of tabular and image datasets and demonstrates the potential to implement several additions to the classic greedy CMI selection policy: these include allowing non-uniform features costs, variable budgets, and incorporating prior information. The results also show that DIME is robust to higher image resolutions, scales to more classes, and benefits from modern architectures. Future work may focus on promising applications like MRIs and region-of-interest selection within WSIs, using DIME to initialize RL methods, and otherwise accelerating or improving DIME's training.

## ACKNOWLEDGEMENT

We thank the members of the Lee Lab for helpful discussions. This work was supported by the National Science Foundation (CAREER DBI-1552309 and 683 DBI-1759487) and the National Institutes of Health (R35 GM 128638 and R01 AG061132).

## REPRODUCIBILITY STATEMENT

The code to train DIME has been submitted as part of the supplementary material. It includes a README that describes the steps to be taken for training and evaluating models with the different datasets. Pseudocode for DIME's training algorithm along with details about our training implementation are provided in Appendix C. Proofs for Lemma 1 and Theorem 1 are provided in Appendix A.1. The proof for Theorem 2 is provided in Appendix A.2. The adaptation of DIME for regression problems, along with the corresponding theorem and proof, are provided in Appendix A.3. Descriptions of all datasets used in our experiments, along with ways to access them, are provided in Appendix D. All the datasets are publicly available except for the intubation dataset, which is private due to patient privacy concerns.

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

# A PROOFS

In this section, we re-state and prove our results from the main text.

## A.1 ESTIMATING CONDITIONAL MUTUAL INFORMATION

**Lemma 1.** *When we use the Bayes classifier $p(\mathbf{y} \mid \mathbf{x}_S)$ as a predictor and $\ell$ is cross entropy loss, the incremental loss improvement is an unbiased estimator of the CMI for each $(x_S, \mathbf{x}_i)$ pair:*

$$\mathbb{E}_{\mathbf{y}, \mathbf{x}_i \mid x_S} [\Delta(x_S, \mathbf{x}_i, \mathbf{y})] = I(\mathbf{y}; \mathbf{x}_i \mid x_S).$$

*Proof.* Consider the expected cross entropy loss given the prediction $p(\mathbf{y} \mid x_S)$:

$$
\begin{aligned}
\mathbb{E}_{\mathbf{y} \mid x_S}[\ell(p(\mathbf{y} \mid x_S), \mathbf{y})] &= -\sum_{y=1}^{K} p(\mathbf{y} = y \mid x_S) \log p(\mathbf{y} = y \mid x_S) \\
&= H(\mathbf{y} \mid x_S).
\end{aligned}
$$

Next, consider the loss given the prediction $p(\mathbf{y} \mid x_S, x_i)$, taken in expectation across $\mathbf{x}_i$ and $\mathbf{y}$'s conditional distribution $p(\mathbf{y}, \mathbf{x}_i \mid x_S)$:

$$
\begin{aligned}
\mathbb{E}_{\mathbf{y}, \mathbf{x}_i \mid x_S}[\ell(p(\mathbf{y} \mid x_S, \mathbf{x}_i), \mathbf{y})] &= \mathbb{E}_{\mathbf{x}_i \mid x_S} \mathbb{E}_{\mathbf{y} \mid x_S, \mathbf{x}_i = x_i}[\ell(p(\mathbf{y} \mid x_S, \mathbf{x}_i = x_i), \mathbf{y})] \\
&= \mathbb{E}_{\mathbf{x}_i \mid x_S}[H(\mathbf{y} \mid x_S, \mathbf{x}_i = x_i)] \\
&= H(\mathbf{y} \mid x_i, \mathbf{x}_i).
\end{aligned}
$$

We therefore have the following expectation for the incremental loss improvement:

$$
\begin{aligned}
\mathbb{E}_{\mathbf{y}, \mathbf{x}_i \mid x_S}[\Delta(x_S, \mathbf{x}_i, \mathbf{y})] &= \mathbb{E}_{\mathbf{y}, \mathbf{x}_i \mid x_S}[\ell(p(\mathbf{y} \mid x_S), \mathbf{y}) - \ell(p(\mathbf{y} \mid x_S, \mathbf{x}_i), \mathbf{y})] \\
&= H(\mathbf{y} \mid x_S) - H(\mathbf{y} \mid x_S, \mathbf{x}_i) \\
&= I(\mathbf{y}; \mathbf{x}_i \mid x_S).
\end{aligned}
$$

Thus, the loss improvement $\Delta(x_S, \mathbf{x}_i, \mathbf{y})$ is an unbiased estimator of the CMI $I(\mathbf{y}; \mathbf{x}_i \mid x_S)$. $\quad\square$

**Theorem 1.** *When $\ell$ is cross entropy loss, the objectives eq. (2) and eq. (3) are jointly optimized by a predictor $f(x_S; \theta^*) = p(\mathbf{y} \mid x_S)$ and value network where $v_i(x_S; \phi^*) = I(\mathbf{y}; \mathbf{x}_i \mid x_S)$ for $i \in [d]$.*

*Proof.* Similar to (Covert et al., 2023), our proof considers both models' optimal predictions for each input. Beginning with the predictor, consider the output given the input $x_S$. The selections were made given only $x_S$, so observing this input conveys no information about $\mathbf{y}$ or the remaining features $\mathbf{x}_{[d] \setminus S}$. The expected loss is therefore

$$\mathbb{E}_{\mathbf{y} \mid x_S}[\ell(f(x_S; \theta), \mathbf{y})].$$

Assuming that $\ell$ is cross entropy loss, we can decompose the expected loss as follows:

$$
\begin{aligned}
\mathbb{E}_{\mathbf{y} \mid x_S}[\ell(f(x_S; \theta), \mathbf{y})] &= \sum_{y=1}^{K} p(\mathbf{y} = y \mid x_S) \log f_y(x_S; \theta) \\
&= \sum_{y=1}^{K} p(\mathbf{y} = y \mid x_S) \log p(\mathbf{y} = y \mid x_S) \frac{f_y(x_S; \theta)}{p(\mathbf{y} = y \mid x_S)} \\
&= H(\mathbf{y} \mid x_S) + D_{\mathrm{KL}}(p(\mathbf{y} \mid x_S) \mid\mid f(x_S; \theta)).
\end{aligned}
$$

Due to the non-negative KL divergence term, we see that the optimal prediction is $p(\mathbf{y} \mid x_S)$. We can make this argument for any input $x_S$, so we say that the optimal predictor is $f(x_S; \theta^*) = p(\mathbf{y} \mid x_S)$ for all $x_S$. Notably, this argument does not depend on the selection policy: it only requires that the policy has no additional information about the response variable or unobserved features.

Next, we consider the value network while assuming that we use the optimal predictor $f(\mathbf{x}_S; \theta^*)$. Given an input $x_S$, we once again have no further information about $\mathbf{y}$ or $\mathbf{x}_{[d] \backslash S}$, so the expected loss is taken across the distribution $p(\mathbf{y}, \mathbf{x}_i \mid x_S)$ as follows:

$$\mathbb{E}_{\mathbf{y}, \mathbf{x}_i \mid x_S} \left[ (v(x_S; \phi) - \Delta(x_S, \mathbf{x}_i, \mathbf{y}))^2 \right].$$

The expected loss can then be decomposed,

$$\mathbb{E}_{\mathbf{y}, \mathbf{x}_i \mid x_S} \left[ (v(x_S; \phi) - \Delta(x_S, \mathbf{x}_i, \mathbf{y}))^2 \right] = \mathbb{E}_{\mathbf{y}, \mathbf{x}_i \mid x_S} \left[ (v(x_S; \phi) - \mathbb{E}_{\mathbf{y}, \mathbf{x}_i \mid x_S} [\Delta(x_S, \mathbf{x}_i, \mathbf{y})])^2 \right]$$
$$+ \operatorname{Var} \left( \Delta(x_S, \mathbf{x}_i, \mathbf{y})) \mid x_S \right),$$

which reveals that the optimal prediction is $\mathbb{E}_{\mathbf{y}, \mathbf{x}_i \mid x_S} [\Delta(x_S, \mathbf{x}_i, \mathbf{y})]$. Following Lemma 1, we know that this is equal to $I(\mathbf{y}; \mathbf{x}_i \mid x_S)$. And because we can make this argument for any $x_S$, we conclude that the optimal value network is given by $v(x_S; \phi^*) = I(\mathbf{y}; \mathbf{x}_i \mid x_S)$. □

## A.2 PRIOR INFORMATION

Before proving Theorem 2, we first present a preliminary result analogous to Lemma 1.

**Lemma 2.** *When we use the Bayes classifier $p(\mathbf{y} \mid \mathbf{x}_S, \mathbf{z})$ as a predictor and $\ell$ is cross entropy loss, the incremental loss improvement is an unbiased estimator of the CMI for each $(x_S, z, \mathbf{x}_i)$ tuple:*
$$\mathbb{E}_{\mathbf{y}, \mathbf{x}_i \mid x_S} [\Delta(x_S, \mathbf{x}_i, z, \mathbf{y})] = I(\mathbf{y}; \mathbf{x}_i \mid x_S, z).$$

*Proof.* The proof follows the same logic as Lemma 1, where we consider the expected loss before and after incorporating the additional feature $\mathbf{x}_i$. The only difference is that each expectation must also condition on $\mathbf{z} = z$, so the terms to analyze are

$$\mathbb{E}_{\mathbf{y} \mid x_S, z} [\ell(p(\mathbf{y} \mid x_S, z), \mathbf{y})]$$
$$\mathbb{E}_{\mathbf{y}, \mathbf{x}_i \mid x_S, z} [\ell(p(\mathbf{y} \mid x_S, z, \mathbf{x}_i), \mathbf{y})].$$
□

We now prove the main result for incorporating prior information.

**Theorem 2.** *When $\ell$ is cross entropy loss, the objectives in eq. (5) are jointly optimized by a predictor $f(x_S, z; \theta^*) = p(\mathbf{y} \mid x_S, z)$ and value network where $v_i(x_S, z; \phi^*) = I(\mathbf{y}; \mathbf{x}_i \mid x_S, z)$ for all $i \in [d]$.*

*Proof.* The proof follows the same logic as Theorem 1. For the predictor with input $x_S$, we can decompose the expected loss as follows:

$$\mathbb{E}_{\mathbf{y} \mid x_S, z} [\ell(f(x_S, z; \theta), \mathbf{y})] = H(\mathbf{y} \mid x_S, z) + D_{\mathrm{KL}}(p(\mathbf{y} \mid x_S, z) \mid\mid f(x_S, z; \theta)).$$

This shows that the optimal predictor is $f(x_S, z; \theta^*) = p(\mathbf{y} \mid x_S, z)$. Next, assuming we use the optimal predictor, the value network's expected loss can be decomposed as follows:

$$\mathbb{E}_{\mathbf{y}, \mathbf{x}_i \mid x_S, z} \left[ (v(x_S, z; \phi) - \Delta(x_S, \mathbf{x}_i, z, \mathbf{y}))^2 \right] = \mathbb{E}_{\mathbf{y}, \mathbf{x}_i \mid x_S, z} \left[ (v(x_S, z; \phi) - \mathbb{E}_{\mathbf{y}, \mathbf{x}_i \mid x_S, z} [\Delta(x_S, \mathbf{x}_i, z, \mathbf{y})])^2 \right]$$
$$+ \operatorname{Var} \left( \Delta(x_S, \mathbf{x}_i, z, \mathbf{y})) \mid x_S, z \right).$$

Based on this, Lemma 2 implies that the optimal value network is $v(x_S, z; \phi^*) = I(\mathbf{y}; \mathbf{x}_i \mid x_S, z)$.
□

## A.3 REGRESSION VERSION

Before proving our main result for regression models, we first present a preliminary result analogous to Lemma 1.

**Lemma 3.** *When we use the conditional expectation* $\mathbb{E}[\mathbf{y} \mid \mathbf{x}_S]$ *as a predictor and* $\ell$ *is mean squared error, the incremental loss improvement is an unbiased estimator of the expected reduction in conditional variance for each* $(x_S, \mathbf{x}_i)$ *pair:*

$$
\begin{aligned}
\mathbb{E}_{\mathbf{y}, \mathbf{x}_i \mid x_S}\left[\Delta(x_S, \mathbf{x}_i, \mathbf{y})\right] &= \mathrm{Var}(\mathbf{y} \mid x_S) - \mathbb{E}_{\mathbf{x}_i \mid x_S}[\mathrm{Var}(\mathbf{y} \mid x_S, \mathbf{x}_i)] \\
&= \mathrm{Var}(\mathbb{E}[\mathbf{y} \mid x_S, \mathbf{x}_i] \mid x_S).
\end{aligned}
$$

*Proof.* Consider the expected loss given the prediction $\mathbb{E}[\mathbf{y} \mid x_S]$:

$$
\mathbb{E}_{\mathbf{y} \mid x_S}\left[\left(\mathbb{E}[\mathbf{y} \mid x_S] - \mathbf{y}\right)^2\right] = \mathrm{Var}(\mathbf{y} \mid x_S).
$$

Next, consider the loss given the prediction $\mathbb{E}[\mathbf{y} \mid x_S, x_i]$, taken in expectation across $\mathbf{x}_i$ and $\mathbf{y}$'s conditional distribution $p(\mathbf{y}, \mathbf{x}_i \mid x_S)$:

$$
\begin{aligned}
\mathbb{E}_{\mathbf{y}, \mathbf{x}_i \mid x_S}\left[\left(\mathbb{E}[\mathbf{y} \mid x_S, \mathbf{x}_i] - \mathbf{y}\right)^2\right] &= \mathbb{E}_{\mathbf{x}_i \mid x_S} \mathbb{E}_{\mathbf{y} \mid x_S, \mathbf{x}_i = x_i}\left[\left(\mathbb{E}[\mathbf{y} \mid x_S, \mathbf{x}_i] - \mathbf{y}\right)^2\right] \\
&= \mathbb{E}_{\mathbf{x}_i \mid x_S}[\mathrm{Var}(\mathbf{y} \mid x_S, \mathbf{x}_i)].
\end{aligned}
$$

We therefore have the following expectation for the incremental loss improvement:

$$
\mathbb{E}_{\mathbf{y}, \mathbf{x}_i \mid x_S}[\Delta(x_S, \mathbf{x}_i, \mathbf{y})] = \mathrm{Var}(\mathbf{y} \mid x_S) - \mathbb{E}_{\mathbf{x}_i \mid x_S}[\mathrm{Var}(\mathbf{y} \mid x_S, \mathbf{x}_i)].
$$

Using the law of total variance, we can simplify this difference as follows:

$$
\mathbb{E}_{\mathbf{y}, \mathbf{x}_i \mid x_S}[\Delta(x_S, \mathbf{x}_i, \mathbf{y})] = \mathrm{Var}\left(\mathbb{E}[\mathbf{y} \mid x_S, \mathbf{x}_i] \mid x_S\right).
$$

This provides a measure similar to the CMI: it quantifies to what extent different plausible values of $\mathbf{x}_i$ affect our best estimate for the response variable. $\qquad\square$

We now present our main result for regression models.

**Theorem 3.** *When* $\ell$ *is mean squared error, the objectives eq. (2) and eq. (3) are jointly optimized by a predictor* $f(x_S; \theta^*) = \mathbb{E}[\mathbf{y} \mid x_S]$ *and value network where for* $i \in [d]$ *we have*

$$
v(x_S; \phi^*) = \mathrm{Var}(\mathbb{E}[\mathbf{y} \mid x_S, \mathbf{x}_i] \mid x_S).
$$

*Proof.* We follow the same proof technique as in Theorem 1. The expected loss for the predictor with input $x_S$ can be decomposed as follows,

$$
\begin{aligned}
\mathbb{E}_{\mathbf{y} \mid x_S}[\ell(f(x_S; \theta), \mathbf{y})] &= \mathbb{E}_{\mathbf{y} \mid x_S}\left[\left(f(x_S; \theta) - \mathbf{y}\right)^2\right] \\
&= \mathbb{E}_{\mathbf{y} \mid x_S}\left[\left(f(x_S; \theta) - \mathbb{E}[\mathbf{y} \mid x_S]\right)^2\right] + \mathrm{Var}(\mathbf{y} \mid x_S),
\end{aligned}
$$

which shows that the optimal predictor network is $f(x_S; \theta^*) = \mathbb{E}[\mathbf{y} \mid x_S]$. Assuming we use the optimal predictor, the expected loss for the value network can then be decomposed as

$$\mathbb{E}_{\mathbf{y},\mathbf{x}_i|x_S} \left[ (v(x_S; \phi) - \Delta(x_S, \mathbf{x}_i, \mathbf{y}))^2 \right] = \mathbb{E}_{\mathbf{y},\mathbf{x}_i|x_S} \left[ (v(x_S; \phi) - \mathbb{E}_{\mathbf{y},\mathbf{x}_i|x_S}[\Delta(x_S, \mathbf{x}_i, \mathbf{y})])^2 \right] \\ + \mathrm{Var}\left( \Delta(x_S, \mathbf{x}_i, \mathbf{y}) \right) \mid x_S ),$$

which shows that the optimal value network prediction is $\mathbb{E}_{\mathbf{y},\mathbf{x}_i|x_S}[\Delta(x_S, \mathbf{x}_i, \mathbf{y})]$. Lemma 3 lets us conclude that the optimal value network is therefore $v(x_S; \phi^*) = \mathrm{Var}(\mathbb{E}[\mathbf{y} \mid x_S, \mathbf{x}_i] \mid x_S)$. □

### A.4 ALLOWING A VARIABLE FEATURE BUDGET

We first re-state our informal claim, and then introduce notation required to show a formal version.

**Proposition 1.** *(Informal) For any feature budget $k$, the best policy to achieve this budget on average achieves lower loss than the best policy with a per-prediction budget constraint. Similarly, for any confidence level $m$, the best policy to achieve this confidence on average achieves lower cost than the best policy with a per-prediction confidence constraint.*

In order to account for a policy's stopping criterion, we generalize our earlier notation so that policies are functions of the form $\pi(x_S) \in \{0\} \cup [d]$, and we say a policy terminates (or stops selecting new features) when $\pi(x_S) = 0$. Given an input $x$, we let $S(\pi, x) \subseteq [d]$ denote the set of indices selected upon termination. The cost for this prediction is $c(\pi, x) = \sum_{i \in S(\pi,x)} c_i$, and there is also a notion of expected loss $\ell(\pi, x)$ that we define as follows:

$$\ell(\pi, x) = \mathbb{E}_{\mathbf{y}|x_{S(\pi,x)}}[\ell(f(x_{S(\pi,x)}), \mathbf{y})]. \tag{6}$$

For example, if $\ell$ is cross entropy loss and we use the Bayes classifier $f(x_S) = p(\mathbf{y} \mid x_S)$, we have $\ell(\pi, x) = H(\mathbf{y} \mid x_S)$; due to this interpretation of the expected loss, we refer to constraints on $\ell(\pi, x)$ as *confidence constraints*. For example, Chattopadhyay et al. (2023) suggests selecting features until $H(\mathbf{y} \mid x_S) \leq m$ for a confidence level $m$. In comparing policies, we must consider the tradeoff between accuracy and feature cost, and we have two competing objectives – the average loss and the average cost:

$$\ell(\pi) = \mathbb{E}[\ell(\pi, \mathbf{x})] \qquad c(\pi) = \mathbb{E}[c(\pi, \mathbf{x})]. \tag{7}$$

Now, there are three types of policies we wish to compare: (1) those that adopt a budget constraint for each prediction, (2) those that adopt a confidence constraint for each prediction, and (3) those with no constraints. These classes of selection policies are defined as follows:

1. (Budget-constrained) These policies adopt a budget $k$ that must be respected for each input $x$. That is, we have $c(\pi, x) \leq k$ for all $x$. This can be ensured by terminating the policy when the budget is exactly satisfied (Ma et al., 2019; Rangrej and Clark, 2021; Covert et al., 2023) or when there are no more candidates that will not exceed the budget. Policies of this form are said to belong to the set $\Pi_k$.

2. (Confidence constrained) These policies adopt a minimum confidence $m$ that must be respected for each input $m$. That is, we must have $\ell(\pi, x) \leq m$ for all $x$. Technically, we may not be able to guarantee this for all predictions due to inherent uncertainty, so we can instead keep making predictions as long as the expected loss exceeds $m$ (Chattopadhyay et al., 2023). Policies of this form are said to belong to the set $\Pi_m$.

3. (Unconstrained) These policies have no per-prediction constraints on the feature cost or expected loss. These are said to belong to the set $\Pi$, where we have $\Pi_k \subseteq \Pi$ and $\Pi_m \subseteq \Pi$.

With these definitions in place, we now present a more formal version of our claim.

**Proposition 2.** *(Formal) For any average feature cost $k$, the best unconstrained policy achieves lower expected loss than the best budget-constrained policy:*

$$\min_{\pi \in \Pi : c(\pi) \leq k} \ell(\pi) \leq \min_{\pi \in \Pi_k} \ell(\pi). \tag{8}$$

*Similarly for any average confidence level $m$, the best unconstrained policy achieves lower expected cost than the best confidence-constrained policy:*

$$\min_{\pi \in \Pi : \ell(\pi) \leq m} c(\pi) \leq \min_{\pi \in \Pi_m} c(\pi). \tag{9}$$

In other words, for any desired average feature cost or confidence level, it cannot help to adopt that level as a per-prediction constraint. The best policy to achieve these levels *on average* can violate the constraint for some predictions, and as a result provide either a lower average cost or expected loss.

*Proof.* The proof of this claim relies on the fact that $\Pi_k \subseteq \Pi$ and $\Pi_m \subseteq \Pi$. It is easy to see that $\Pi_k \subseteq \{\pi \in \Pi : c(\pi) \leq k\}$. This implies the inequality in eq. (8) because the right-hand side takes the minimum over a smaller set of policies. Similarly, it is easy to see that $\Pi_m \subseteq \{\pi \in \Pi : \ell(\pi) \leq m\}$, which implies the inequality in eq. (9). □

## B    PREDICTOR SUBOPTIMALITY

Consider a feature subset $x_S$ where the ideal prediction from the Bayes classifier is $p(\mathbf{y} \mid x_S)$, but the learned classifier instead outputs $q(\mathbf{y} \mid x_S)$. The incorrect prediction can result in a skewed loss, which then provides incorrect labels to the value network $v(x_S; \phi)$. Specifically, the expected loss assuming many data points $(\mathbf{x}, \mathbf{y})$ such that $\mathbf{x}_S = x_S$ becomes

$$\mathbb{E}_{\mathbf{y}|x_S}[\ell(q(\mathbf{y} \mid x_S), \mathbf{y})] = H(\mathbf{y} \mid x_S) + D_{\text{KL}}\left(p(\mathbf{y} \mid x_S) \parallel q(\mathbf{y} \mid x_S)\right). \tag{10}$$

The loss is therefore higher on average than it should be given the Bayes classifier, with the extra loss being equal to the KL divergence between the ideal and actual predictions. However, this does not imply that $v(x_S; \phi)$ systematically overestimates the CMI, because its labels are based on the expected loss *reduction*.

Consider that the above situation with incorrect predictions occurs not only for $x_S$, but also for all values of $\mathbf{x}_i$: that is, the classifier outputs $q(\mathbf{y} \mid x_S, x_i)$ rather than $p(\mathbf{y} \mid x_S, x_i)$ for each value $x_i$. Now, the expected loss reduction is the following:

$$\mathbb{E}_{\mathbf{y}, \mathbf{x}_i | x_S}[\Delta(x_S, \mathbf{x}_i, \mathbf{y})] = I(\mathbf{y}; \mathbf{x}_i \mid x_S) + D_{\text{KL}}(p(\mathbf{y} \mid x_S) \parallel q(\mathbf{y} \mid x_S))$$
$$- \mathbb{E}_{\mathbf{x}_i | x_S}\left[D_{\text{KL}}(p(\mathbf{y} \mid x_S, \mathbf{x}_i) \parallel q(\mathbf{y} \mid x_S, \mathbf{x}_i))\right]. \tag{11}$$

This implies that given infinite data and a value network that perfectly optimizes its objective, the learned CMI estimates are biased according to a *difference* in KL divergence terms. Notably, the difference can be either positive or negative, so the CMI estimates can be incorrect in either direction. And intuitively, the bias shrinks to zero as the classifier approaches $p(\mathbf{y} \mid x_S)$ for all predictions.

## C  TRAINING ALGORITHM

Algorithm 1 summarizes our learning approach, where we jointly train the predictor and value networks according to the objectives in eq. (2) and eq. (3). We implemented it in PyTorch (Paszke et al., 2017) using PyTorch Lightning.[3] Note that the algorithm requires a dataset of fully observed $\mathbf{x}$ samples with corresponding labels $\mathbf{y}$.

---

**Algorithm 1:** Training algorithm

---

**Input:** Data distribution $p(\mathbf{x}, \mathbf{y})$, learning rate $\gamma$, budget $k$, exploration $\epsilon \in (0, 1)$, costs $c \in \mathbb{R}_+^d$
**Output:** Predictor $f(\mathbf{x}_S; \theta)$, value network $v(\mathbf{x}_S; \phi)$

```
// Prepare models
```
initialize $v(\mathbf{x}_S; \phi)$, pre-train $f(\mathbf{x}_S; \theta)$ with random masks

**while** *not converged* **do**
    ```// Initialize variables```
    initialize $S = \{\}, \mathcal{L}_\theta = 0, \mathcal{L}_\phi = 0$
    sample $x, y \sim p(\mathbf{x}, \mathbf{y})$

    ```// Initial prediction```
    calculate $\hat{y}_{\text{prev}} = f(x_{\{\}}; \theta)$
    update $\mathcal{L}_\theta \leftarrow \mathcal{L}_\theta + \ell(\hat{y}_{\text{prev}}, y)$

    **while** $\sum_{i \in S} c_i \leq k$ **do**
        ```// Determine next selection```
        calculate $I = v(x_S; \phi)$
        set $j = \arg\max_{i \notin S} I_i / c_i$ with probability $1 - \epsilon$, else sample $j$ from $[d] \setminus S$

        ```// Update predictor loss```
        update $S \leftarrow S \cup j$
        calculate $\hat{y} = f(x_S; \theta)$
        update $\mathcal{L}_\theta \leftarrow \mathcal{L}_\theta + \ell(\hat{y}, y)$

        ```// Update value loss```
        calculate $\Delta = \ell(\hat{y}_{\text{prev}}, y) - \ell(\hat{y}, y)$
        update $\mathcal{L}_\phi \leftarrow \mathcal{L}_\phi + (I_j - \Delta)^2$
        set $\hat{y}_{\text{prev}} = \hat{y}$
    **end**

    ```// Gradient step```
    update $\theta \leftarrow \theta - \gamma \nabla_\theta \mathcal{L}_\theta, \quad \phi \leftarrow \phi - \gamma \nabla_\phi \mathcal{L}_\phi$
**end**

---

[3]https://www.pytorchlightning.ai

Next, Algorithm 2 shows how features are selected at inference time. For simplicity, we show only the penalized stopping criterion, which requires a penalty term $\lambda > 0$. To implement either the budget or confidence constraints described in Section 4.3, we only need to change the stopping criterion.

---

**Algorithm 2:** Inference algorithm

---

**Input:** Test instance $(x, y)$, predictor $f(\mathbf{x}_S; \theta)$, value network $v(\mathbf{x}_S; \phi)$, penalty parameter $\lambda > 0$
**Output:** Prediction $\hat{y}$

```
// Initialize feature set
```
initialize $S = \{\}$

**for** $i = 1, \ldots, d$ **do**
```
    // Estimate current CMI
```
 calculate $I = v(x_S; \phi)$

```
    // Check stopping criterion
```
 **if** $\max_{i \notin S} I_i / c_i < \lambda$ **then**
  | break
 **end**

```
    // Determine next selection
```
 set $j = \arg\max_{i \notin S} I_i / c_i$
 update $S \leftarrow S \cup j$

**end**

```
// Make prediction
```
calculate $\hat{y} = f(x_S; \theta)$

---

To incorporate prior information into Algorithm 1 and Algorithm 2 (denoted by $\mathbf{z}$), we can simply update $I = v(x_S, z; \phi)$ and $\hat{y} = f(x_S, z; \theta)$ during both training and inference.

Algorithm 1 is simplified to omit several details that we implement in practice, and these details are discussed below.

**Masked pre-training.** When pre-training the predictor $f(\mathbf{x}_S; \theta)$, we sample feature subsets as follows: we first sample a cardinality $\{0, \ldots, d\}$ uniformly at random, and we then sample the members of the subset at random. This distribution ensures even coverage of different subset sizes $|S|$, whereas treating each feature's membership as an independent Bernoulli variable biases the subsets towards a specific size.

**Minibatching.** As is conventional in deep learning, we calculate gradients in parallel for multiple inputs. In Algorithm 1, this means that we take gradient steps calculated over multiple data samples $(\mathbf{x}, \mathbf{y})$ and multiple feature budgets.

**Learning rate schedule.** Rather than train with a fixed learning rate $\gamma > 0$, we reduce its value over the course of training. To avoid setting a precise number of epochs for each dataset, we decay the learning rate when the loss reaches a plateau, and we perform early stopping when the learning rate is sufficiently low. The initial learning rate depends on the architecture, but we use values similar to those used for conventional training (e.g., ViTs require a lower learning rate than CNNs or MLPs).

**Annealing exploration probability.** Setting a large value for $\epsilon$ helps encourage exploration, but at inference time we set $\epsilon = 0$. To avoid the mismatch between these settings, we anneal $\epsilon$ towards zero over the course of training. Specifically, we train the model with a sequence of $\epsilon$ values, warm-starting each model with the output from the previous value.

**Parameter sharing.** As mentioned in Section 4, we sometimes share parameters between the predictor and value network. We implement this via a shared backbone, e.g., a sequence of self-attention layers in a ViT (Dosovitskiy et al., 2020). The backbone is initialized via the predictor pre-training with random masks, and it is then used for both $f(\mathbf{x}_S; \theta)$ and $v(\mathbf{x}_S; \phi)$ with separate output heads for each one.

**Scaling value network outputs.** To learn the optimal value network outputs, it is technically sufficient to let the network make unconstrained, real-valued predictions. However, given that the true CMI values are non-negative, or $I(\mathbf{y}; \mathbf{x}_i \mid x_S) \geq 0$ for all $(x_S, \mathbf{x}_i)$, it is sensible to constrain the predictions: for example, we can apply a softplus output activation. Similarly, we know that the true CMI values are upper bounded by the current prediction entropy $H(\mathbf{y} \mid x_S)$ (Cover and Thomas, 2012). These simultaneous bounds can be summarized as follows:

$$0 \leq I(\mathbf{y}; \mathbf{x}_i \mid x_S) \leq H(\mathbf{y} \mid x_S).$$

To enforce both inequalities, we apply a sigmoid operation to the unconstrained value network prediction $v(x_S; \phi)$, and we multiply this by the empirical prediction entropy from $f(x_S; \theta)$. An ablation showing the effect of this approach is in Figure 9.

**Prior information.** We found that an issue with using prior information (as discussed in Section 4.2) is overfitting to $\mathbf{z}$. This is perhaps unsurprising, particularly when $\mathbf{z}$ is high-dimensional, because the same input is used repeatedly with different feature subsets $\mathbf{x}_S$ and the same label $\mathbf{y}$. To mitigate this, we applied the following simple fix: for the separate network that processes the prior variable $\mathbf{z}$, we detached gradients when using the learned representation to make classifier predictions, so that gradients are propagated only for the value network's CMI predictions. An ablation demonstrating this approach is in Figure 15.

**Inference time.** At inference time, we follow a similar procedure as in Algorithm 1 but with $\epsilon = 0$, so that we always make the most valuable selection. Algorithm 2 demonstrates the pseudo-code for inference. In terms of stopping criteria for making a prediction, we explore multiple approaches, as discussed in Section 4.3: (1) a budget-constrained approach with parameter $k$, (2) a confidence constrained approach with parameter $m$, and (3) a penalized approach with parameter $\lambda$. Our results are generated by evaluating a single learned policy with several values for each of these parameters. The range of reasonable values for the confidence parameter $m$ and penalty parameter $\lambda$ depend on the dataset, so these are tuned by hand.

**Feature grouping.** Several of our datasets involve grouped features: for example, we group pixels in the image datasets into patches, and our medical diagnosis datasets have grouped one-hot indicators for categorical variables (Intubation, ROSMAP). To implement this grouping structure in our method, we simply predict the CMI for each group, and then calculate the value network's objective based on the loss improvement after revealing the group's values.

## D  DATASETS

This section provides details about the datasets used in our experiments. The size of each dataset is summarized in Table 1.

Table 1: Summary of datasets used in our experiments.

| Dataset | # Features | # Feature Groups | # Classes | # Samples |
|---|---|---|---|---|
| MNIST | 784 | – | 10 | 60,000 |
| Intubation | 112 | 35 | 2 | 65,515 |
| ROSMAP | 46 | 43 | 2 | 13,438 |
| ImageNette | 50,176 | 196 | 10 | 13,395 |
| ImageNet-100 | 50,176 | 196 | 100 | 135,000 |
| Histopathology | 50,176 | 196 | 2 | 3152 |

**MNIST.** This is the standard digit classification dataset (LeCun et al., 1998). We downloaded it with PyTorch and used the standard train and test splits, with $10,000$ training samples held out as a validation set.

**Intubation.** This is a privately curated dataset from a university medical center, gathered over a 13-year period (2007-2020). Our goal is to predict which patients require respiratory support upon arrival in the emergency department. We selected 112 pre-hospital clinical features including dispatch information (injury date, time, cause, and location), demographic information (age, sex),

and pre-hospital vital signs (blood pressure, heart rate, respiratory rate). The outcome is defined based on whether a patient received respiratory support, including both invasive (intubation) and non-invasive (BiPap) approaches. We excluded patients under the age of 18, and because many features represent one-hot encodings for categorical variables, we grouped them into 35 feature groups. Feature acquisition costs were obtained by having a board-certified emergency physician estimate the relative cost of obtaining each feature. The dataset is not publicly available due to patient privacy concerns.

**ROSMAP.** The Religious Order Study (ROS) and Memory Aging Project (MAP) (A Bennett et al., 2012a;b) are complementary epidemiological studies that enroll participants to study dementia. ROS is a logitudinal study that enrolls clergy without known dementia from across the United States, including Catholic nuns, priests, and brothers aged 65 years and older. Participants agree to annual medical and psychological evaluation and pledge their brain for donation. MAP is a longitudinal study that enrolls participants encompassing a wider community from 40 continuous care retirement facilities around the Chicago metropolitan area. Participants are without known dementia and agree to annual clinical evaluation and donation of brain, spinal cord and muscle after death. While entering the study, participants share demographic information (e.g. age, sex) and also provide their blood samples for genotyping. At each annual visit, their medical information is updated and they take a series of cognitive tests, which generate multiple measurements over time. This results in 46 different variables, grouped into 43 feature groups to account for one-hot encodings. The task is to predict dementia onset within the next three years given the current medical information and no prior history of dementia. In total, the data contains 3,194 individuals with between 1 and 23 annual visits. Following the preprocessing steps used in (Beebe-Wang et al., 2021), we applied a four-year sliding window over each sample, thereby generating multiple samples per participant. Each sample is split into an input window consisting of the current year visit $t$ and a prediction window of the next three years $(t + 1, t + 2, t + 3)$. To avoid overlap between the training, validation, or testing sets, we ensured that all samples from a single individual fell into only one of the data splits. Feature acquisition costs expressed in terms of time taken were borrowed from (Beebe-Wang et al., 2021) for the cognitive tests and rough estimates were assigned to the remaining features using prior knowledge. We discarded the genotypic feature (APOE e4 allele) from the feature set since it is highly predictive of dementia and it is difficult to assign an appropriate cost. The dataset can be accessed at `https://dss.niagads.org/cohorts/religious-orders-study-memory-and-aging-project-rosmap/`.

**Imagenette and ImageNet-100.** These are both subsets of the standard ImageNet dataset (Deng et al., 2009). Imagenette contains 10 classes and was downloaded using the Fast.ai deep learning library (Howard), ImageNet-100 contains 100 classes and was downloaded from Kaggle (Ambityga), and in both cases we split the images to obtain train, validation and test splits. Images were resized to $224 \times 224$ resolution for both architectures we explored, ResNets (He et al., 2016b) and ViTs (Dosovitskiy et al., 2020).

**MHIST.** The MHIST (**m**inimalist **hist**opathology) (Wei et al., 2021) dataset comprises 3,152 hematoxylin and eosin (H&E)-stained Formalin Fixed Paraffin-Embedded (FFPE) fixed-size images of colorectal polyps from patients at Dartmouth-Hitchcock Medical Center (DHMC). The task is to perform binary classification between hyperplastic polyps (HPs) and sessile serrated adenomas (SSAs), which is a challenging prediction task with significant variation in inter-pathologist agreement (Abdeljawad et al., 2015; Farris et al., 2008; Glatz et al., 2007; Khalid et al., 2009; Wong et al., 2009). HPs are typically benign, while SSAs are precancerous lesions that can turn into cancer if not treated promptly. The fixed-size images were obtained by scanning 328 whole-slide images and then extracting regions of size $224 \times 224$ representing diagnostically-relevant regions of interest for HPs or SSAs. For the ground truth, each image was assigned a gold-standard label determined by the majority vote of seven board-certified gastrointestinal pathologists at the DHMC. The dataset can be accessed by filling out the form at `https://bmirds.github.io/MHIST/`.

# E   MODELS

Here, we briefly describe the types of models used for each dataset. The exploration probability $\epsilon$ for all models is set to $0.05$ at the start with an annealing rate of $0.2$.

**Tabular datasets.** For all the tabular datasets, we use multilayer perceptrons (MLPs) with two hidden layers and ReLU non-linearity. We use $128$ neurons in the hidden layers for the ROSMAP and Intubation datasets, and $512$ neurons for MNIST. The initial learning rate is set to $10^{-3}$ at the start and we also use dropout with probability $0.3$ in all layers to reduce overfitting (Srivastava et al., 2014). The value and predictor networks use separate but identical network architectures. The networks are trained on a NVIDIA RTX 2080 Ti GPU with 12GB of memory.

**Image datasets: ResNet.** We use a shared ResNet-50 backbone (He et al., 2016b) for the predictor and value networks. The final representation from the backbone has shape $7 \times 7$, and the output heads for each network are specified as follows. The predictor head contains a Conv $\rightarrow$ Batch Norm $\rightarrow$ ReLU sequence followed by global average pooling and a fully connected layer. The value network head consists of an upsampling block with a transposed convolutional layer, followed by a $1 \times 1$ convolution and a sigmoid to scale the predictions (see Appendix C). The learning rate starts at $10^{-5}$, and the networks are trained on a NVIDIA RTX 2080 Ti GPU with 12GB of memory.

**Image datasets: ViT.** We use a shared ViT backbone (`vit_small_patch16_224`) (Dosovitskiy et al., 2020) for the predictor and value networks. We use ViT and ResNet backbones having a similar number of parameters for a fair comparison: ResNet-50 has approximately 23M parameters, and ViT-Small has 22M parameters. The predictor head contains a linear layer applied to the class token, and the value network head contains a linear layer applied to all tokens except for the class token, followed by a sigmoid function. When incorporating prior information, a separate ViT backbone is used for both the predictor and value networks to generate an embedding, which is then concatenated with the masked image embedding to get either the predicted CMIs or the class prediction. The learning rate starts at $10^{-5}$, and the networks are trained on a NVIDIA Quadro RTX 6000 GPU with 24GB of memory.

# F   BASELINES

Here, we provide more details here on our baseline methods.

**Concrete autoencoder.** This is a static feature selection method that optimizes a differentiable selection module within a neural network (Balın et al., 2019). The layer can be added at the input of any architecture, so we use this method for both tabular and image datasets. The original work suggested training with an exponentially decayed temperature and a hand-tuned number of epochs, but we use a different approach to minimize the tuning required for each dataset: we train with a sequence of temperature values, and we perform early stopping for each one based on the validation loss. We return the features that are selected after training with the lowest temperature, and we evaluate them by training a model from scratch with only those features provided.

**CMICOT.** This is a static feature selection method that scores features based on their mutual information with the response variable. To identify joint interactions between several features, the authors build a scoring function using a min-max optimization objective (Shishkin et al., 2016). To make the method practically feasible, binary representations of features are used. To adapt this method to our setting, for each feature budget, we use CMICOT to select the best subset and then fit a classifier on the selected features to get the final performance. We ensure that the classifier used has the same architecture as the one used for DIME.

**mRMR.** This is a static feature selection method that identifies a subset of features from a larger set that maximizes the relevance to the target variable while minimizing redundancy among selected features (Peng et al., 2005). In other words, it aims to find a set of features that are individually informative for predicting the target variable and, at the same time, not highly correlated with each other. Similar to CMICOT, we fit a separate classifier on the subset identified for each feature budget.

**EDDI.** This is a DFS method that uses a generative model to sample the unobserved features (Ma et al., 2019). We implement a PVAE to sample the unknown features, and these samples are used to estimate the CMI for candidate features at each selection step. We separately implement a classifier

that makes predictions with arbitrary feature sets, similar to the one obtained after masked pre-training in Algorithm 1. We use this method only for our tabular datasets, as the PVAE is not expected to work well for images, and the computational cost at inference time is relatively high due to its iteration across candidate features.

**Probabilistic hard attention.** This method extends EDDI to work for images by imputing unobserved features within a low-dimensional, learned feature space (Rangrej and Clark, 2021). To ensure that the method operates on the same image regions as DIME, we implemented a feature extractor that separately computes embeddings for non-overlapping $14 \times 14$ patches, similar to a ViT (Dosovitskiy et al., 2020) or bag-of-features model (Brendel and Bethge, 2019). Specifically, our extractor consists of a $16 \times 16$ convolutional layer, followed by a series of $1 \times 1$ convolutions. The features from each patch are aggregated by a recurrent module, and we retain the same structure used in the original implementation.

**Argmax Direct.** This is a DFS method that directly estimates the feature index with maximum CMI. It is based on two concurrent works whose main difference is how gradients are calculated for the selector network (Chattopadhyay et al., 2023; Covert et al., 2023); for simplicity, we only use the technique based on the Concrete distribution from Covert et al. (2023). As a discriminative method, this baseline allows us to use arbitrary architectures and is straightforward to apply with either tabular or image datasets.

**Classification with costly features (CwCF).** This is an RL-based approach that converts the DFS problem into a MDP, considering the expected utility of acquiring a feature given its cost and impact on the classification accuracy (Janisch et al., 2019). A variant of deep Q-learning is implemented as the RL solver, and the method is adapted to work with sparse training datasets with missing features. We used the budget-constrained implementation, we added support for feature grouping, and we use the same architectures as DIME for the Q-network. We train and evaluate separate models for each target budget. We limit comparison to tabular datasets because the method does not scale well and high performance is not expected for images. CwCF initially suffered in our evaluation with the medical diagnosis datasets because we use AUROC to measure performance: CwCF produces hard classifications rather than predicted probabilities, so it can suffer on a ranking-based metrics like AUROC that are more meaningful for clinical prediction tasks. To account for this discrepancy, we followed an approach from Erion et al. (2022) and used the Q-values for classes as proxies for pseudo predicted probabilities, since the Q-values for a "class" action can be interpreted as a score of how confident the model is in predicting that class. This resulted in improved performance for the ROSMAP and the Intubation datasets. However, we still observe that CwCF underperforms compared to the other methods.

**Opportunistic learning (OL).** This is another RL-based approach to solve DFS (Kachuee et al., 2018). The model consists of two networks: a Q-network that estimates the value associated with each action, where actions correspond to features, and a P-network responsible for making predictions. These are similar to the value and predictors network used in DIME, so we use the same architectures as our approach, and OL shares parameters between the P- and Q-networks. We use the implementation from (Covert et al., 2023), which modifies the method by preventing the prediction action until the pre-specified budget is met, and supports pre-defined feature groups. We limit comparison to tabular datasets since the method does not scale well and high performance is not expected for images.

## G  ADDITIONAL RESULTS

Here, we present the results of several additional experiments and ablations on the different datasets.

Figure 7 shows the prediction calibration of the predictor network by plotting DIME's performance at different levels of confidence for a specific budget of $k = 15$, along with the density of the samples at those confidence levels across multiple datasets. This shows that the predictor network is well calibrated and does not systematically overestimate or underestimate its predicted probabilities. Proper calibration is important to achieve accurate loss values, because these are then used to train the value network (see eq. (3)).

Figure 8 shows the calibration of the predicted CMIs by the value network for both a tabular and an image dataset by plotting the difference in entropy or losses against the predicted CMI. A linear trend

showcases that the CMIs predicted by the value network align well with the difference in either the entropy or the loss. Since we do not have ground truth CMI values to evaluate the accuracy of our value network, this serves a viable alternative for real-world datasets, and we can verify that the CMI predictions correctly represent the expected reduction in either loss or prediction entropy.

Figure 9 shows the effect of constraining the predicted CMIs using the current prediction entropy, as described in Appendix C. Without the constraint, there are some samples with unrealistically high CMI estimates that are greater than the prediction entropy. After applying the sigmoid activation on the value network predictions, this issue is corrected.

Figure 10 shows multiple trials for the penalized policy on the tabular datasets. This provides a simple way to represent variability between trials when we cannot precisely control the budget between separately trained policies. Similarly, Figure 12 and Figure 13 show multiple trials while considering non-uniform feature costs, and Figure 11 shows multiple trials for the image datasets. The relative results stay the same across all trials, as the performance variability between trials is generally small.

Figure 14 shows the confidence distribution of full input predictions in the tabular datasets. Across all datasets, we observe that the model has high confidence in many of the samples, but there are some that remain uncertain even after observing all the features. This provides motivation for using the penalized approach, because a confidence-constrained approach could suffer here by expending the entire feature acquisition budget only to remain at high uncertainty.

Figure 15 shows the effectiveness of detaching gradients for the predictor network of the sketch $\mathbf{z}$ in the histopathology dataset, as described in Appendix C. The penalized policy performs significantly better when we propagate gradients only for CMI predictions, which we attribute to reduced overfitting to the prior information.

Figure 16 compares the penalized stopping criterion with the confidence- and budget-constrained versions for the image datasets. Similarly, Figures 10 and 11 include the budget-constrained approach in comparisons with the baselines. The penalized stopping criterion introduced in Section 4.3 consistently achieves the best results.

Table 2 compares the training times (in hours) of DIME with the baselines. It is difficult to compare exact training times across methods because each has hyperparameters that can be tuned to make it converge faster, but we compared them under the hyperparameters used to generate our results. We observe that DIME trains slightly faster than Argmax Direct for both the tabular and image datasets. Compared to EDDI and Hard Attention, we see that these generative methods are actually somewhat faster to train; however, because they must generate a large number of samples at inference time and iterate over candidate features, their evaluation time is far slower. The total training and inference time for EDDI is 48.26 hours with MNIST, and for Hard Attn is 25.45 hours on Imagenette, whereas the evaluation time for DIME is negligible compared to its training. As expected, the RL method CwCF is very slow to run on MNIST, taking almost 29 hours to train models for all target budgets.

Finally, Figure 17 shows example feature selections for the MNIST dataset, and Figure 18 shows feature selection frequencies for ROSMAP, to verify the distinct selections made across predictions. This capability differs from static selection methods like the CAE (Balın et al., 2019), which select the same features for all predictions.

Table 2: Training times for each method (in hours).

| Method | MNIST | Imagenette |
|---|---|---|
| DIME | 4.88 | 27.14 |
| Argmax Direct | 6.02 | 43.67 |
| EDDI | 0.39 | - |
| Hard Attn | - | 18.25 |
| CwCF | 28.91 | - |

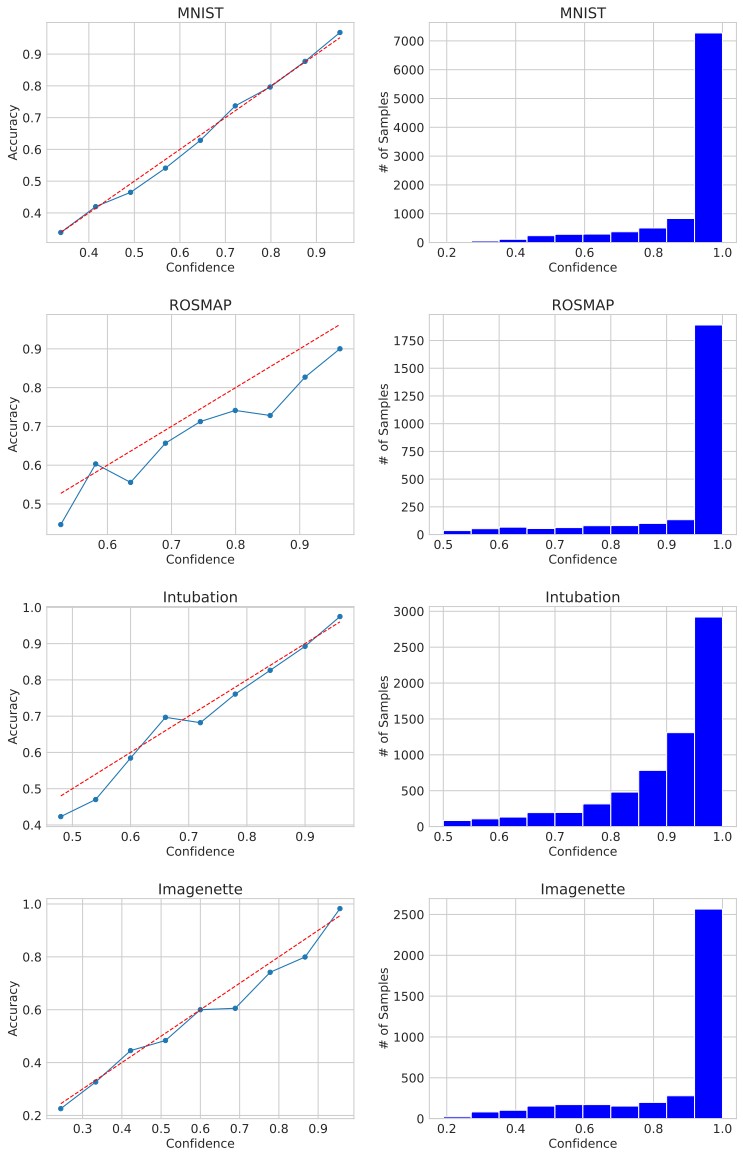

Figure 7: Evaluation of prediction calibration for a fixed budget of $k = 15$. The left column shows the prediction calibration of the predictor network by plotting the accuracy for different confidence levels. The right column shows the distribution of confidences across all samples.

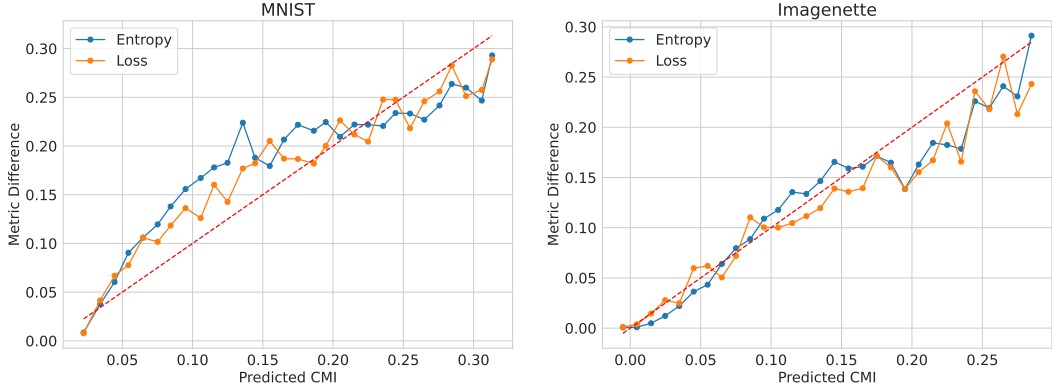

Figure 8: Evaluation of CMI calibration. The x-axis shows different values for the predicted CMI throughout the selection process, and the y-axis shows the reduction in either loss or entropy after the corresponding feature is selected.

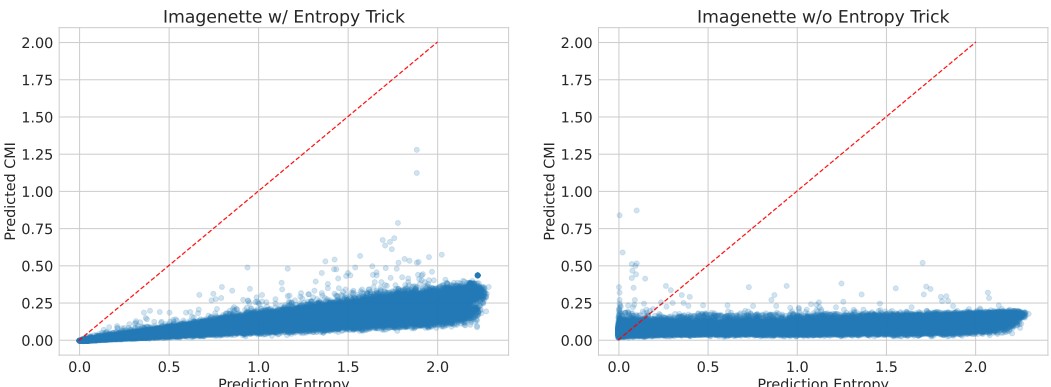

Figure 9: Predicted CMIs with and without the entropy trick to scale value network outputs.

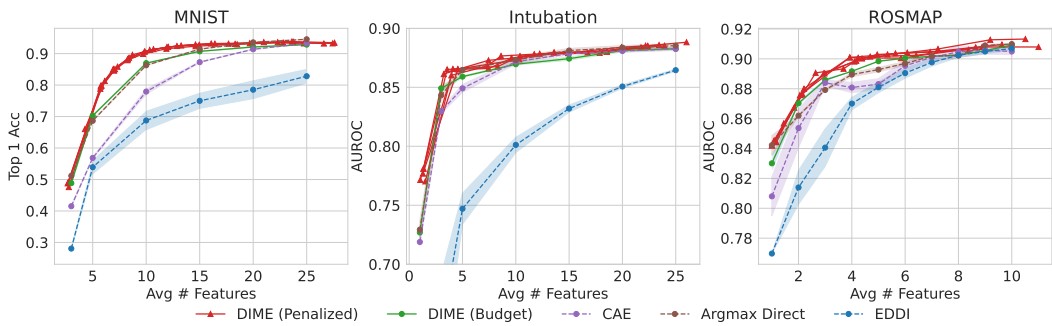

Figure 10: Multiple trials using the penalized policy and the budget constraint for tabular datasets. DIME with penalized policy remains the best method across five independent trials.

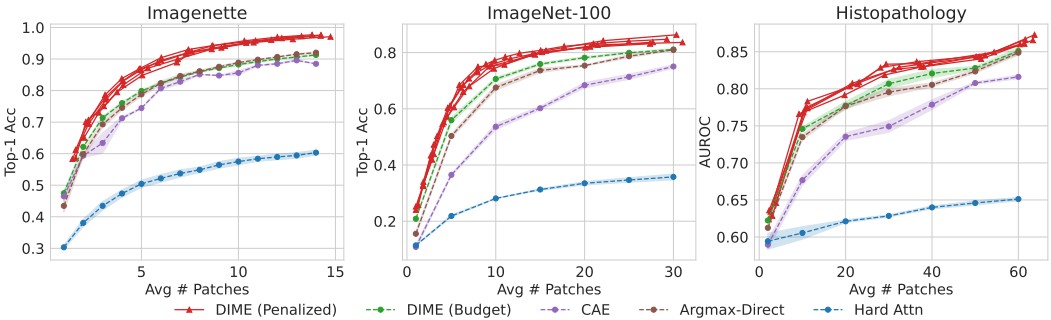

Figure 11: Multiple trials using DIME's penalized policy and the budget constraint for image datasets. DIME with penalized policy remains the best method across five independent trials.

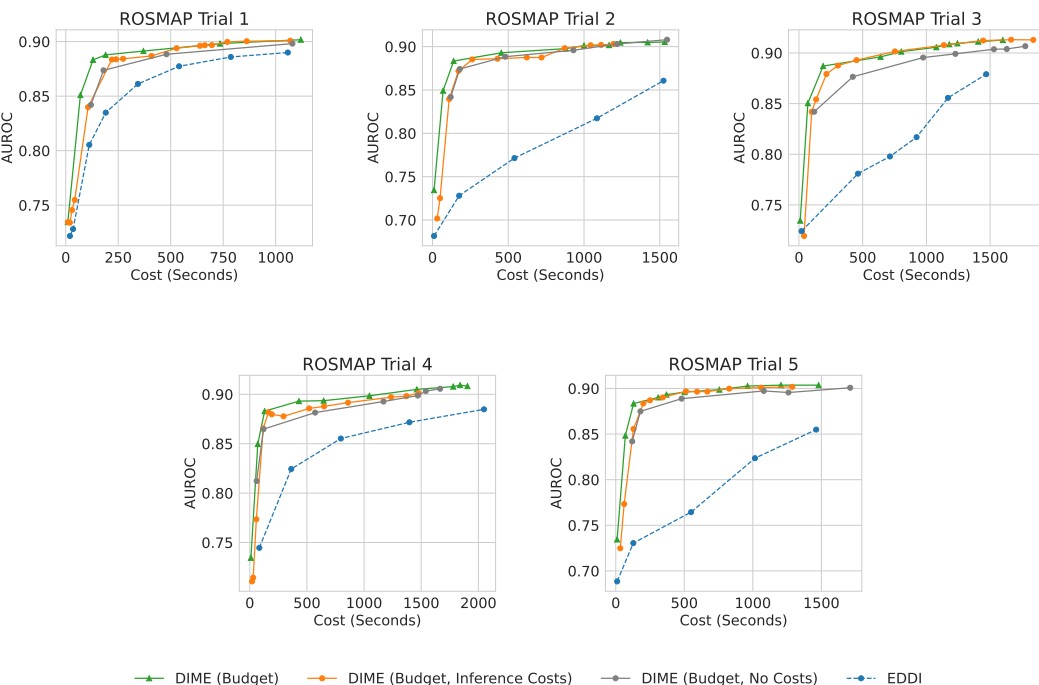

Figure 12: Multiple trials when using non-uniform feature costs for the ROSMAP dataset.

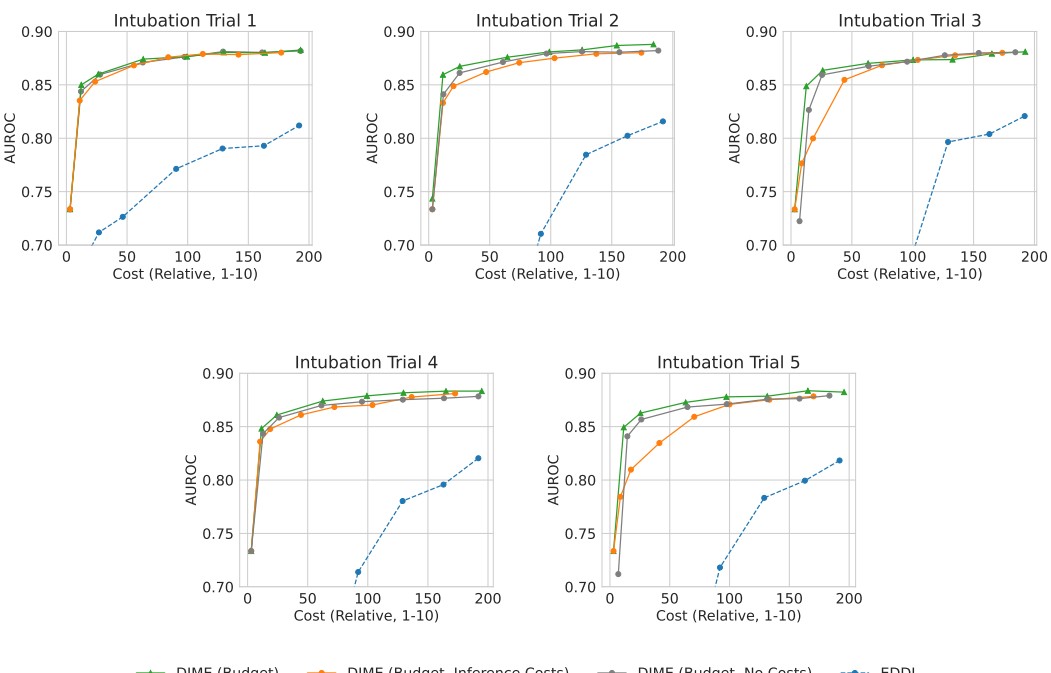

Figure 13: Multiple trials when using non-uniform feature costs for the intubation dataset.

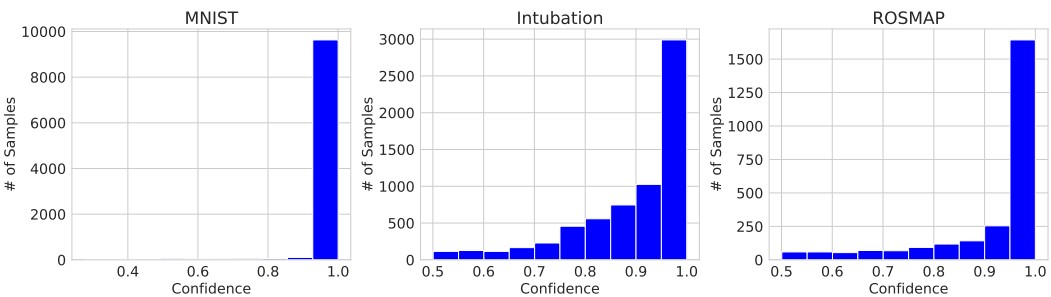

Figure 14: Confidence distribution on full-input predictions across the tabular datasets.

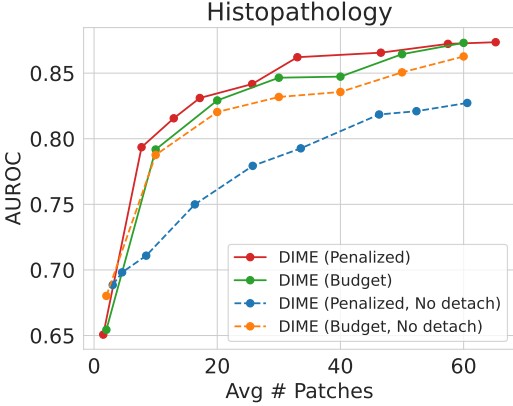

Figure 15: Ablation of stop-gradients trick when using prior information for the histopathology dataset (Appendix C).

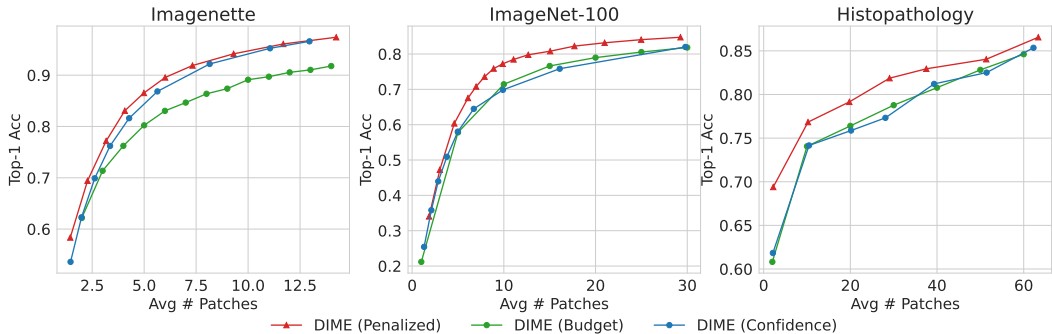

Figure 16: Comparison of the budget-constrained, confidence-constrained and penalized stopping criteria.

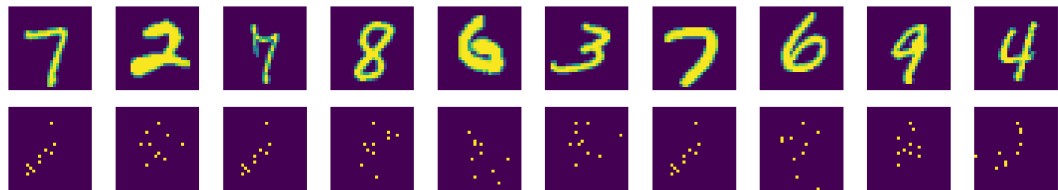

Figure 17: Examples of MNIST feature selections across several samples, with budget $k = 10$.

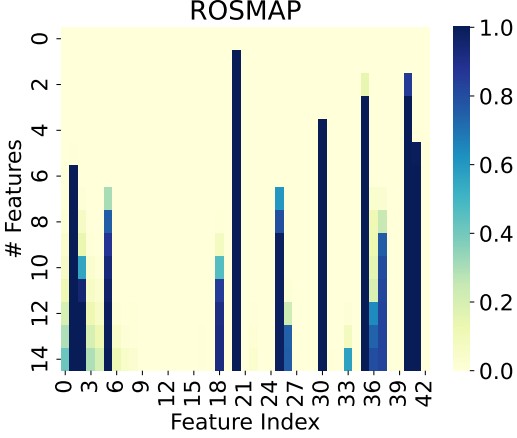

Figure 18: Feature selection frequency for ROSMAP. Each entry shows the fraction of samples in which a feature is selected when we use the budget-constrained stopping criterion for the specified number of features.

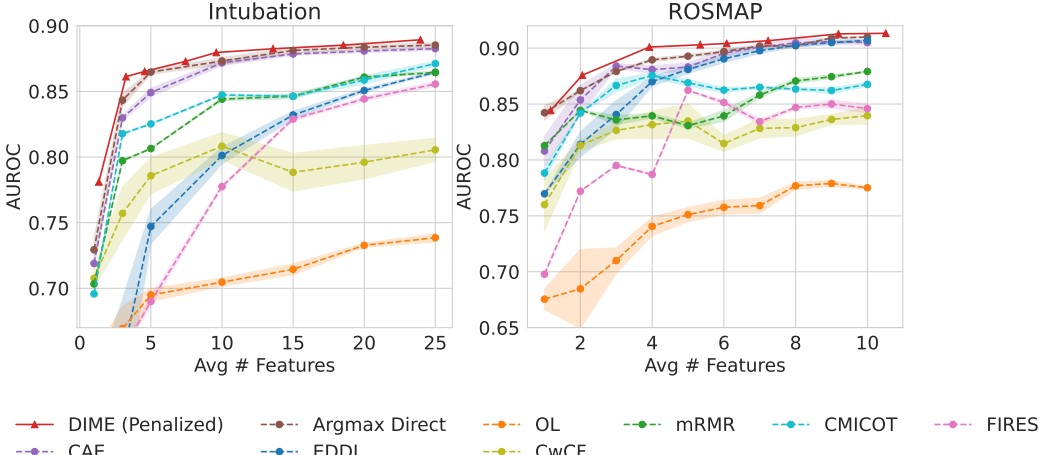

Figure 19: Evaluation with tabular datasets for varying feature acquisition budgets. Results are averaged across 5 trials, and shaded regions indicate the standard error for each method.

