# OpenReview forum: "Estimating Conditional Mutual Information for Dynamic Feature Selection"
_ICLR.cc/2024/Conference — ICLR 2024 poster_

### Official Review · Reviewer_s6Fj · 2023-10-21

**Soundness:** 3 good
**Presentation:** 2 fair
**Contribution:** 3 good
**Rating:** 8
**Confidence:** 4

**Summary:**

The authors propose DIME a dynamic feature selection (DFS) method that is based on two neural networks, aiming to maximize prediction accuracy while balancing the cost of acquiring features. Specifically, a value network is designed to estimate the conditional mutual information (CMI), $I(y;x_i∣x_S)$, and a prediction network is used to make predictions based on the currently selected features. Both networks are jointly trained. If the acquisition costs for features are known, the objective aims to select the feature that maximizes $I(y;x_i∣x_S)/c_i$, where $c_i$ is the acquisition cost of $i$'th feature. In the experiments, DIME is compared against both static and dynamic feature selection methods on various (tabular and image) datasets. The results show that DIME outperforms the competitors in terms of predictive performance and effectiveness (i.e., number of selected features).

**Strengths:**

1.) The research investigates various policies for feature selection, such as RL, imitation learning, and a greedy policy based on CMI.

2.) The introduction of the Predictor Network and Value Network, which jointly work to estimate Conditional Mutual Information (CMI) is interesting and represents a novel approach.

3.) The theoretical parts of the paper provide a solid foundation for the approach.

4.) The experimental evaluation is quite comprehensive.

**Weaknesses:**

1.) While the incorporation of acquisition costs with CMI estimates is straight-forward, the acquisition costs appear to be an artificial supplementary add-on to the methodology.

2.) Some related approaches should be mentioned in the related work section and could have been considered as competitors

Incremental permutation feature importance (iPFI): towards online explanations on data streams (Machine Learning 2023)
F Fumagalli, M Muschalik, E Hüllermeier, B Hammer

Leveraging model inherent variable importance for stable online feature selection (KDD 2020)
J Haug, M Pawelczyk, K Broelemann, G Kasneci

3.) It is not quite clear how prior information is factored into and affects the feature selection policy; this part should be better explained.

4.) An exact algorithmic representation should be used to provide clarity on the interplay between the networks, the CMI estimation, the use of prior information, and how the features are finally selected in various scenarios.

**Questions:**

See weaknesses 3.) and 4.) from above.

**Details Of Ethics Concerns:**

None.

---

> ### Author Response · Authors · 2023-11-15
> **Response to Reviewer s6Fj**
>
> We would like to thank the reviewer for closely examining our work and providing their feedback. We have used the space below to respond to the questions raised in the review.  Updates to the manuscript are highlighted in red to make them easily identifiable.
>
> > “While the incorporation of acquisition costs with CMI estimates is straight-forward, the acquisition costs appear to be an artificial supplementary add-on to the methodology.”
>
> Once we are able to accurately predict the CMI with DIME, extending the method to incorporate non-uniform feature costs may seem like a small add-on, but it makes DIME more useful in real-world settings where features have diverse acquisition costs. Taking the emergency medicine setting as a motivating example, there are many features that can be acquired with significant variation in the temporal and monetary cost. Demographic features like age, sex, and patient history would be cheap and only take only minutes to obtain, whereas lab test features might take days and would be more expensive, and genetic tests could take weeks to conduct. Incorporating non-uniform feature costs allows us to take these differences into consideration, and DIME can use multiple low-cost features in lieu of one high cost feature to get the same diagnostic performance.
>
> > “Some related approaches should be mentioned in the related work section and could have been considered as competitors”
>
> Thank you for pointing us to these very interesting works! We carefully read them both, and after studying their proposed methods, we do not think they are closely related to our problem setting. Fumagalli et al. (2023) study global feature importance in the context of streaming data, where new data points come in and the distribution may change over time; in contrast, we focus on feature selection with selections that can differ between data points, and there is no analogous notion of distribution shift. Haug et al. (2020) focus on online feature selection, which is also related to data streams where the distribution can shift over time; our work considers the iterative acquisition of features for each prediction, but this is very different from the notion of incoming data streams with changing distributions. As such, we do not think there is any room to compare DIME with these methods.
>
> > “It is not quite clear how prior information is factored into and affects the feature selection policy; this part should be better explained”
>
> Section 4.2 describes how our approach can be modified to include prior information: essentially, $z$ becomes an input to both the predictor and value networks, and they are optimized in the same way as before using the same loss functions. The predictor network is trained with the loss $\ell(f(x_S, z; \theta), y)$ (e.g., cross entropy loss) and the value network is trained with $(v_i(x_S, z; \phi) - \Delta(x_S, x_i, z, y))^2$. Theorem 2 describes how this should theoretically affect the dynamic feature selection procedure.
>
> We demonstrate this approach with the histopathology dataset, using a Canny edge image as prior information. During training, we use separate ViT backbones for the original and the edge images, and we pass these images as inputs to both the value network and the predictor. Specifically, we concatenate the resulting embeddings of the original and edge images from the backbones before either getting the CMI estimates from $v$ or the predictions from $f$, which are implemented as small networks that take concatenated embeddings as their input. We can update this section in the manuscript to better explain how the prior information is passed to the networks.
>
> > “An exact algorithmic representation should be used to provide clarity on the interplay between the networks, the CMI estimation, the use of prior information, and how the features are finally selected in various scenarios.”
>
> For more information about how the networks are trained, please see Algorithm 1 in Appendix C, which shows how the value network and predictor network are jointly optimized. We are happy to specify how the algorithm incorporates prior information, as well as add another one describing how features are selected at inference time. We will also add a reference to these algorithms in the main text, which we forgot to include in the original submission.

---

> > ### Comment · Reviewer_s6Fj · 2023-11-15
> > **"... and there is no analogous notion of distribution shift."**
> >
> > Could you please specify/formalize your notion of differences (or assumptions) with respect to the conditional (i.e., concept) distribution P(y|x)? Because if you assume that certain features that are important for one prediction may not be important for another prediction, then you are assuming some kind of differences w.r.t P(y|x) for different data points (e.g., similar to the differences between the distributions in the leaf nodes of a decision tree). If so, then many works concerning feature selection under the assumption of differences w.r.t. the concept distribution become relevant (feature selection under concept drifts would actually be only a small fraction of relevant related works).

---

> > > ### Author Response · Authors · 2023-11-15
> > > **Clarification**
> > >
> > > Sure, we can clarify our assumptions about the data distribution and how they differ from the concept drift setting. We assume that there is just one data distribution $p(\mathbf{x}, \mathbf{y}) = p(\mathbf{x})p(\mathbf{y} \mid \mathbf{x})$ and that all data points are drawn from this distribution during both training and inference. We also assume that for most $x$ values, there exists a feature subset $x_S$ that mostly determines the response variable, or $p(\mathbf{y} \mid x_S) \approx p(\mathbf{y} \mid x)$. Our goal is to discover the small number of features as cheaply as possible and make the corresponding prediction, and DIME does so by modeling $f(x_S ; \theta) \approx p(\mathbf{y} \mid x_S)$ and making selections based on $v_i(x_S; \phi) \approx I(\mathbf{y}; \mathbf{x}_i \mid x_S)$.
> > >
> > > For example, images containing an object will have the object in just one location, and this corresponds to the subset $x_S$ for that image $x$. This does not mean that the response distribution $p(\mathbf{y} | \mathbf{x})$ changes, as we assume that all images are drawn from the same $p(\mathbf{x})$ and all labels from the same $p(\mathbf{y} \mid \mathbf{x})$ (albeit with different $x$ values). That is why we say there is no relevant notion of concept drift in our work.
> > >
> > > For completeness, we experimented with the FIRES codebase to make absolutely sure. Given our full training set, FIRES outputs a single feature ranking that we can use for static feature selection. We could update the ranking as new test instances come in, but this makes little difference because these are samples from the same distribution, and in any case the features cannot adapt to the current inference example $x$ without first observing its label $y$ (which defeats the purpose of per-prediction selections). The static feature selections from FIRES were not competitive with our best static baseline (CAE) on the two medical diagnosis datasets, and we couldn’t run it on MNIST because the public implementation only supports binary classification. The results are now shown in Figure 19 of our updated submission (at the end of the appendix), but as we already have strong static baselines and this usage does not take advantage of what FIRES is designed to do, we think these results are best not to include in the paper.

---

> > > > ### Comment · Reviewer_s6Fj · 2023-11-15
> > > > **Thanks for the clarification!**
> > > >
> > > > Thank you for clarifying your assumptions. I think the work would benefit from highlighting this difference between your approach and feature selection strategies in dynamic data scenarios. Overall, you have addressed my concerns well, and I will raise my score accordingly.

---

> ### Author Response · Authors · 2023-11-16
> **Thank you for the response**
>
> Thank you for taking the time to read our response and adjust your score, we appreciate it! We will consider the differences between our method and data streams.

---

### Official Review · Reviewer_j5dM · 2023-10-27

**Soundness:** 3 good
**Presentation:** 4 excellent
**Contribution:** 2 fair
**Rating:** 6
**Confidence:** 4

**Summary:**

The work is devoted to the problem of dynamic feature selection. The authors propose to estimate – learn – Conditional Mutual Information (CMI) of the next feature given the past selected ones. The learning is based on the incremental loss improvements of a given predictor (simultaneously learned with the CMI). It is proven that, for cross entropy loss, the optimal learning point will be CMI. The paper also contains experiments on several datasets that demonstrate better performance of the proposed method in comparison to several baselines.

**Strengths:**

-	Clear problem setup
-	Large volume of experiments

**Weaknesses:**

-	Comparison with baselines (see questions)
-	It looks like incorporating prior information and variable feature selection constitute 2/3 of the overall work contribution, but they seem more-or-less incremental things (i.e., given such a task a practitioner would easily reproduce Sections 4.2 and 4.3 having Section 4.1). More non-trivial insights would be nice to see in the main contribution.

**Questions:**

1.	As far as I understand, the authors propose a method to learn CMI improvement. It would be nice to compare this method to the ones, where CMI is approximated in the direct statistical way. For instance, there is a lot of approaches like MIM, MIFS, mRMR, JMI, CMICOT, etc. See for a brief overview of them in [Shishkin, A., et al. “Efficient high-order interaction-aware feature selection based on conditional mutual information”, NeurIPS 2016]. Could you please compare your approach with such statistical methods? Including, in experiments. Is it true that the gain we obtain from learning CMI improvement over statistical methods is cost effective? (learning consumes more computation => on the same level of computation costs, we can get more from statistical methods, right?)


2.	Returning back to [Shishkin, A., et al. “Efficient high-order interaction-aware feature selection based on conditional mutual information”, NeurIPS 2016], we can see that greedy approach of selecting features one by one is not the best way. The ideal approach is to select the best subset of features (which is computationally and dimensionally infeasible). But we at least can select features not one-by-one, but taking into account interactions between selecting features (like in CMICOT). Is there any way to incorporate that approach into the work of the authors?

---

> ### Author Response · Authors · 2023-11-15
> **Response to Reviewer j5dM**
>
> We would like to thank the reviewer for closely examining our work and providing their feedback. We have used the space below to respond to the questions raised in the review.  Updates to the manuscript are highlighted in red to make them easily identifiable.
>
> > “As far as I understand, the authors propose a method to learn CMI improvement. It would be nice to compare this method to the ones, where CMI is approximated in the direct statistical way. For instance, there is a lot of approaches like MIM, MIFS, mRMR, JMI, CMICOT, etc. See for a brief overview of them in [Shishkin, A., et al. “Efficient high-order interaction-aware feature selection based on conditional mutual information”, NeurIPS 2016]. Could you please compare your approach with such statistical methods? Including, in experiments. Is it true that the gain we obtain from learning CMI improvement over statistical methods is cost effective? (learning consumes more computation => on the same level of computation costs, we can get more from statistical methods, right?)”
>
> The reviewer is correct that DIME estimates each feature’s CMI with the response variable given a set of observed features. However, our problem formulation is different from all the methods that the reviewer points to: these methods perform *static feature selection*, which means that the features are selected just once and used for all predictions, whereas DIME performs *dynamic feature selection*, which means that we select features separately for each prediction. To be precise, these static methods are concerned with the quantity $I(\mathbf{y}; \mathbf{x}_i \mid \mathbf{x}_S)$, while DIME aims to estimate $I(\mathbf{y}; \mathbf{x}_i \mid x_S)$ with $x_S$ being a set of observed features for the current prediction. Estimating these two quantities requires very different methodologies, and it is not clear if the procedures used by the suggested methods are applicable here or if they are computationally feasible to be run repeatedly for each selection performed for each prediction.
>
> As for the static feature selection methods mentioned by the reviewer, we did not include these initially because we instead use a more recent state-of-the-art static baseline: the Concrete Autoencoder (Balin et al., 2019). For completeness, we tried comparing DIME with mRMR and CMICOT, and the results are now shown in Figure 2. We can confirm that these methods both underperform DIME. CMICOT does better than mRMR, but both methods underperform the Concrete Autoencoder, our original static baseline. Note that we chose mRMR and CMICOT among the suggested methods because CMICOT has been shown to outperform the others and both of them had public implementations that were easy to incorporate into our problem.
>
> > “Returning back to [Shishkin, A., et al. “Efficient high-order interaction-aware feature selection based on conditional mutual information”, NeurIPS 2016], we can see that greedy approach of selecting features one by one is not the best way. The ideal approach is to select the best subset of features (which is computationally and dimensionally infeasible). But we at least can select features not one-by-one, but taking into account interactions between selecting features (like in CMICOT). Is there any way to incorporate that approach into the work of the authors?”
>
> As described in the previous response, a key difference between CMICOT and DIME is whether features are selected just once (static) or separately for each prediction (dynamic). In our case, DIME can only observe features once they are acquired, so features must be selected one at a time. However, in a sense DIME’s training procedure does consider feature interactions, because the CMI conditions on the subset of features that are already observed. For example, if two features are perfectly correlated, after observing one of them, the other’s CMI would drop to zero because acquiring it will provide no additional information about the response variable.

---

> > ### Author Response · Authors · 2023-11-15
> > **Response to Reviewer j5dM (Continued)**
> >
> > > “It looks like incorporating prior information and variable feature selection constitute 2/3 of the overall work contribution, but they seem more-or-less incremental things (i.e., given such a task a practitioner would easily reproduce Sections 4.2 and 4.3 having Section 4.1). More non-trivial insights would be nice to see in the main contribution.”
> >
> > We agree that our work’s most important contribution is the novel discriminative approach for estimating the CMI (Section 4.1). However, the extensions to incorporate prior information and variable feature budgets are important and enabled by the technique in Section 4.1, so they are worth including in our work. Incorporating prior information (Section 4.2) is important in real-world settings where there are “free” features available at the start that can guide the selection of other features, and the penalized stopping criterion (Section 4.3) is an important ingredient for achieving better performance (see for example Figure 4 or Figure 16).
> >
> > Furthermore, these ideas were not exactly trivial to formulate, even if they seem natural following Section 4.1. Please see Appendix C for a gradient stopping trick we developed to prevent overfitting with prior information, and see Appendix A.4 for our proof regarding the suboptimality of a hard budget constraint.

---

> > > ### Author Response · Authors · 2023-11-20
> > > **Any Remaining Concerns**
> > >
> > > Dear Reviewer j5dM,\
> > > The discussion deadline is approaching and we are wondering if you had a chance to look at our response. Please let us know if you have any remaining questions or concerns we can address before the discussion period ends.

---

> > > > ### Author Response · Authors · 2023-11-23
> > > > **Discussion period ending**
> > > >
> > > > Dear reviewer j5dM,
> > > >
> > > > The deadline for the discussion period is later today, so we wanted to check in one last time about our response. The main issue in your review seems to be the relationship with earlier methods that estimate the mutual information in a statistical fashion. As we clarified in our response, the methods you mentioned solve a different problem than our work, because they perform *static* rather than *dynamic* feature selection (i.e., they don't select features separately for each prediction). To the best of our knowledge, these techniques are not straightforward to apply to dynamic feature selection, so they are quite different from DIME and not the most relevant comparisons.
> > > >
> > > > Nonetheless, to address this point in our work, we have 1) added new citations in the related work section about static selection methods, and 2) added two methods you mentioned as additional static baselines (mRMR and CMICOT). The results show that these methods underperform DIME, and they also underperform our previous static baseline (the CAE). Neither result is surprising, because DIME has the advantage of selecting informative features separately for each prediction, and the CAE directly optimizes over the feature set rather than following any heuristics (this is why we initially included it as a strong static baseline).
> > > >
> > > > We hope this resolves your concern, and we would appreciate you updating your review if you have the chance. If you have any other questions or concerns, we would be happy to address them during the remainder of the discussion period.

---

### Official Review · Reviewer_iU6i · 2023-10-31

**Soundness:** 3 good
**Presentation:** 3 good
**Contribution:** 3 good
**Rating:** 8
**Confidence:** 5

**Summary:**

This paper studies the problem of dynamic feature selection (i.e., adaptively acquire features based on previously acquired ones), where the ultimate goal is to classify each data instance using a small subset of the available features. A greedy approach, termed DIME, is proposed, where feature selection is based on the conditional mutual information (CMI) with the response variable. The paper focuses on estimating CMI in a discriminative fashion, and then trains two networks to implement the feature selection policy. Furthermore, the paper discusses how to handle non-uniform costs across features, prior information and variable feature budgets. A number of experiments on tabular and image datasets are provided that showcase the performance of DIME in contrast to prior work on the area of dynamic feature selection.

**Strengths:**

+ The paper studies a very important and timely problem, i.e., dynamic feature selection with a minimal budget.
+ A new greedy algorithm is proposed that estimates CMI in a discriminative fashion.
+ The paper discusses how to handle common issues happening in practice, e.g., non-uniform costs, variable feature budgets, etc.
+ Theoretical results are provided that lead to an unbiased estimator of CMI.
+ The paper is well-written and effort has been put to clarify its contributions.
+ The performance of the proposed algorithm is illustrated on a number of datasets and compared with existing methods.

**Weaknesses:**

- I believe that some relevant wont on dynamic feature selection that is neither RL-based nor greedy-based is missing (see references below)
- The performance improvement of DIME wrt to one of the baselines (i.e., Argmax Direct) is small (especially for the tabular datasets).
- The paper claims that DIME only accounts for non-uniform costs and variable feature budgets, but I am pretty sure that some of the RL-based methods handle such settings too.
- The really bad performance of the RL-based methods compared to the greedy methods, including DIME, is quite surprising to me and I am wondering if this is some kind of typo.

**Questions:**

(1) I believe that the following set of references also study dynamic feature selection, but from a different perspective than the references presented in the paper. Specifically, they formulate the problem as POMDP and study the theoretical properties of the optimum solution, which enables them to design fast algorithms for dynamic feature selection. They also give bounds on the expected number of features needed to achieve a specific accuracy level. Finally, the latter paper uses mutual information to also drive dynamic feature selection. I think that after reviewing these papers, the authors would agree with me that this is relevant work and they should consider citing them for completeness.

- Liyanage, Y.W., Zois, D.S. and Chelmis, C., 2021. Dynamic instance-wise joint feature selection and classification. IEEE Transactions on Artificial Intelligence, 2(2), pp.169-184.

- Liyanage, Y.W., Zois, D.S. and Chelmis, C., 2021. Dynamic Instance-Wise Classification in Correlated Feature Spaces. IEEE Transactions on Artificial Intelligence, 2(6), pp.537-548.

(2) I also believe that the algorithms proposed in the previous references as well as (Janisch et al; 2019) are designed to handle non-uniform costs and variable feature budgets. I think this need to be clearly stated in the paper, since at the moment, it is suggested that only DIME can accommodate such settings.

(3) The performance of DIME is very close to Argmax Direct. What is the benefit of using DIME instead of Argmax Direct in that sense?

(4) My major concern is the really surprisingly bad performance of RL-based methods compared to the greedy methods, including DIME. The reason is that I see the RL-based methods as a way of implementing dynamic programming, which is the optimal thing to do, and should be better than greedy methods. Of course, RL has its own issues, but I am still wondering if the performance improvement observed in the experiments is due to using more powerful neural networks to approximate the selection process and not the greedy nature of the method. In addition, it may be because the RL-based methods that DIME is compared with use different criteria than mutual information (excluding the missing references above). Finally, I believe that the RL-based methods consider the cost of features during their decision making process. Is it possible that the results presented in the paper have misspecified this parameter? May be an ablation study  would help me better justify the above surprising behavior of RL-based methods compared to DIME

(5) I noticed in the Appendix that features are in groups. Does DIME use the group structure somehow?

(6) It is unclear to me from the description in the paper if DIME uses a hard feature budget constraint for each data instance or an average constraint or something else. May be you can clarify this?

---

> ### Author Response · Authors · 2023-11-15
> **Response to Reviewer iU6i**
>
> We would like to thank the reviewer for closely examining our work and providing their feedback. We have used the space below to respond to the questions raised in the review. Updates to the manuscript are highlighted in red to make them easily identifiable.
>
> > “I believe that some relevant work on dynamic feature selection that is neither RL-based nor greedy-based is missing”
>
> We agree that formulating the problem as a POMDP and using a Bayesian network to capture feature dependencies are valid approaches, and we are happy to include other references in the related work. The suggested works are now included in the updated manuscript, with descriptions based on our best understanding of those works.
>
> > “I also believe that the algorithms proposed in the previous references as well as (Janisch et al; 2019) are designed to handle non-uniform costs and variable feature budgets. I think this need to be clearly stated in the paper, since at the moment, it is suggested that only DIME can accommodate such settings”
>
> We don’t wish to suggest that DIME is the only method that allows non-uniform costs and variable feature budgets. In the original submission, we cite Janisch et al. (2019) and Kachuee et al. (2018) as examples of methods that consider these, see for example the following quote: “For example, Janisch et al. (2019) formulate DFS as a MDP where the reward is the 0-1 loss minus the feature cost, while considering both variable budgets and non-uniform feature costs.” Notice that we also compare DIME to EDDI and CwCF in Figure 3 where we consider non-uniform costs.
>
> However, in the context of greedy approaches for dynamic feature selection, it’s worth noting that most previous works focus on the fixed-budget setting with uniform costs. DIME is therefore unique in enabling these additions to the greedy approach, and this is possible only because it more accurately estimates the CMI in a discriminative fashion. If it is helpful, we can update the manuscript to make this point clearer.
>
> > “The performance of DIME is very close to Argmax Direct. What is the benefit of using DIME instead of Argmax Direct in that sense?”
>
> The performance of Argmax Direct approaches that of DIME for two tabular datasets (Intubation and ROSMAP), but for the other 4/6 datasets we tested, DIME substantially outperforms Argmax Direct for almost all feature budgets. Furthermore, DIME addresses real-world challenges that Argmax-Direct cannot address: first, feature acquisition costs are not always identical for all features (e.g., a genetic test takes longer than obtaining demographic features), and second, allowing variable per-sample feature budgets enables quicker diagnosis for certain samples (e.g., it may become immediately clear that a patient with a neck injury needs intubation). Argmax Direct does not estimate the CMI, which means that it cannot be adjusted when using non-uniform feature costs, and the CMI values cannot be incorporated into a stopping criterion (Section 4.3). By estimating the CMI with DIME, we are able to do both of these; the former is crucial to support for some applications, and the latter leads to clear performance improvements under a given average budget. To help readers appreciate these points, we will emphasize them in our revised conclusion.

---

> > ### Author Response · Authors · 2023-11-15
> > **Response to Reviewer iU6i (Continued)**
> >
> > > “The really bad performance of the RL-based methods compared to the greedy methods, including DIME, is quite surprising to me and I am wondering if this is some kind of typo.”
> >
> > We agree that RL theoretically has the capacity to discover better policies than a greedy approach, but there has been prior work showing that it does not perform as well in practice, so our paper is not unique in this finding. See for example [1, 2, 3]: [1] uses CwCF (Janisch et al; 2019) as one of the baselines and it consistently underperforms other methods, similar to our work. [2] uses a different RL method (RAM) and finds that it is not competitive with greedy methods. [3] compares Argmax Direct with OL and similar underperforming results are observed. We use implementations from the official GitHub repositories for both RL methods that we compare to, and we ensure that the network architectures are identical to DIME. We also use the same hyperparameters from the official implementations.
> >
> > One of the reasons for CwCF’s underperformance could be that we use AUROC rather than accuracy for the clinical datasets. Since CwCF produces hard classifications rather than predicted probabilities, it might suffer on ranking-based metrics like AUROC that are more meaningful for clinical prediction tasks. We see that CwCF is more competitive on the MNIST dataset which uses accuracy as the evaluation metric. To account for this discrepancy, we followed the approach from [1] and used the Q-values for classes as proxies for predicted probabilities since the Q-values for a “class” action can be interpreted as a score of how confident the model is in that class. Using Q-values resulted in improved performance for the ROSMAP and the Intubation datasets, and we have updated Figures 2 and 3 in the manuscript using this new setup. However, it is still observed that CwCF underperforms the other methods. We thank the reviewer for bringing this to our attention and we hope that the updated figures clarify the original concern. The OL baseline uses a separate P-network to make predictions, so it would not suffer from the same issue.
> >
> > > “I noticed in the Appendix that features are in groups. Does DIME use the group structure somehow?”
> >
> > Yes, several of our datasets involve grouped features: for example, we group pixels in the image datasets into patches, and our medical diagnosis datasets have grouped one-hot indicators for categorical variables (Intubation, ROSMAP). This grouping structure is easy to implement in our method, because we can simply predict the CMI for each group, and then calculate the value network’s objective based on the loss improvement after revealing the group’s values. We now discuss this in more detail in Appendix C.
> >
> > > “It is unclear to me from the description in the paper if DIME uses a hard feature budget constraint for each data instance or an average constraint or something else. Maybe you can clarify this?”
> >
> > The **Variable budgets** part of Section 4.3 describes the stopping criterion we use, and it’s an important modification that results in improved performance. A hard budget constraint is suboptimal for achieving the best performance, so we design an alternative that accounts for how much remaining information is expected based on the CMI estimates. We adopt a penalty parameter $\lambda > 0$, then make selections at each step with $\text{argmax}_i I(y; x_i | x_S) / c_i$, and we stop collecting features when $\max_i I(y; x_i | x_S) / c_i < \lambda$. Thus, there is no hard budget constraint for each sample. We call this a *penalized stopping criterion* and use it when comparing to the baselines for all the datasets (Figures 2, 3, 5). Each point on the DIME curves corresponds to a $\lambda$ value that obtains the corresponding average number of features across all samples. We also compared the penalized stopping criterion to other possible criteria, namely budget-constrained and confidence-constrained (Supplementary Figure 16), and we observe improved performance with the penalized criterion.
> >
> > [1] Erion et al. "A cost-aware framework for the development of AI models for healthcare applications." Nature Biomedical Engineering (2022).
> >
> > [2] Chattopadhyay et al. "Variational information pursuit for interpretable predictions." International Conference on Learning Representations (2023).
> >
> > [3] Covert et al. "Learning to maximize mutual information for dynamic feature selection." International Conference on Machine Learning (2023).

---

> > > ### Author Response · Authors · 2023-11-20
> > > **Any Remaining Concerns**
> > >
> > > Dear Reviewer iU6i,\
> > > The discussion deadline is approaching and we are wondering if you had a chance to look at our response. Please let us know if you have any remaining questions or concerns we can address before the discussion period ends.

---

> > > > ### Comment · Reviewer_iU6i · 2023-11-21
> > > >
> > > > The response has addressed most of my comments and I will be raising my score. Nonetheless, I have left a few more comments to be addressed.

---

> > > ### Comment · Reviewer_iU6i · 2023-11-21
> > >
> > > Thanks for the clarifications. They have clarified many of the concerns that I had. Regarding the RL vs greedy question, you mentioned about the use of AUROC vs accuracy. I am still wondering if this is not the only reason. As I mentioned earlier, RL-based methods such as CwCF (Janisch et al; 2019) make decisions based on expected misclassification cost and the feature acquisition cost, but DIME uses mutual information to guide feature acquisition. This should play some role, right? In any case, I feel that the Introduction of the paper strongly suggests that RL-based methods are not good alternatives for the problem studied in the paper. However, it seems that there is limited work to suggest that in addition to the fact that it is not for sure that RL is worse, since there are many factors that can influence the final result (e.g., the objective a method optimizes or even the datasets selected for the experiments). I suggest that the Introduction be modified to clarify this point.

---

> > ### Comment · Reviewer_iU6i · 2023-11-21
> >
> > Thanks for your response and the clarifications. I believe it will be beneficial to also include in the revised paper the discussion about handling non-uniform costs and variable feature budgets.

---

> ### Author Response · Authors · 2023-11-21
> **Thank you for response**
>
> Thank you for your response and for engaging in the discussion period! We appreciate your comments and have made a couple additional changes to the introduction.
>
> 1. We now specify that greedy methods have been found to outperform RL methods in several specific works, rather than making a more general claim. We also mention what we believe is the main reason RL methods currently don't work as well: policy optimization is a much more difficult learning problem. In the DFS context, RL incurs sparse rewards and must explore an exponential state space, so even with an appropriate reward like classification accuracy it's understandably difficult to train. With DIME, learning the greedy CMI policy involves fitting one classifier model (the predictor network) and one regression model (the value network), and training these is nearly as simple as standard supervised learning.
>
> 2. We added further clarification that some existing RL methods consider non-uniform costs and variable feature budgets (Janisch et al, Kachuee et al). We added this in the discussion of our contributions, and we clarified that these capabilities are new for greedy CMI-based methods.
>
> Thanks again for engaging in the discussion, and we appreciate you adjusting your score!

---

> > ### Comment · Reviewer_iU6i · 2023-11-23
> >
> > Thank you

---

### Meta-Review · Area_Chair_1Jxo · 2023-12-06

**Metareview:**

The authors study dynamic feature selection with the ultimate goal of classifying each data instance using a small subset of the available features. They propose a greedy approach (called DIME) based on the conditional mutual information. They suggest learning this conditional mutual information by a discriminative approach.  Overall the reviewers found the work interesting but they raised concerned about the poor performance of the method compared to RL-based ones.

**Justification For Why Not Higher Score:**

I do not appreciate this work, but the scores are quite high.

**Justification For Why Not Lower Score:**

I do not feel comfortable overturning the reviewers, but personally find the work underwhelming and shallow.

---

### Decision · Program_Chairs · 2024-01-16

Accept (poster)